# Fast deep neural correspondence for tracking and identifying neurons in *C. elegans* using semi-synthetic training

Xinwei Yu[1], Matthew S Creamer[2], Francesco Randi[1], Anuj K Sharma[1], Scott W Linderman[3,4], Andrew M Leifer[1,2]*

[1]Department of Physics, Princeton University, Princeton, United States; [2]Princeton Neuroscience Institute, Princeton University, Princeton, United States; [3]Department of Statistics, Stanford University, Stanford, United States; [4]Wu Tsai Neurosciences Institute, Stanford University, Stanford, United States

**Abstract** We present an automated method to track and identify neurons in *C. elegans*, called 'fast Deep Neural Correspondence' or fDNC, based on the transformer network architecture. The model is trained once on empirically derived semi-synthetic data and then predicts neural correspondence across held-out real animals. The same pre-trained model both tracks neurons across time and identifies corresponding neurons across individuals. Performance is evaluated against hand-annotated datasets, including NeuroPAL (Yemini et al., 2021). Using only position information, the method achieves 79.1% accuracy at tracking neurons within an individual and 64.1% accuracy at identifying neurons across individuals. Accuracy at identifying neurons across individuals is even higher (78.2%) when the model is applied to a dataset published by another group (Chaudhary et al., 2021). Accuracy reaches 74.7% on our dataset when using color information from NeuroPAL. Unlike previous methods, fDNC does not require straightening or transforming the animal into a canonical coordinate system. The method is fast and predicts correspondence in 10 ms making it suitable for future real-time applications.

*For correspondence:
leifer@princeton.edu

Competing interests: The authors declare that no competing interests exist.

## Introduction

The nervous system of the nematode *C. elegans* is well characterized, such that each of the 302 neurons is named and has stereotyped locations across animals (*White et al., 1986*; *Sulston, 1976*; *Witvliet et al., 2020*). The capability to find corresponding neurons across animals is essential to investigate neural coding and neural dynamics across animals. Despite the worm's overall stereotypy, the variability in neurons' spatial arrangement is sufficient to make predicting neural correspondence a challenge. For whole-brain calcium imaging (*Schrödel et al., 2013*; *Venkatachalam et al., 2016*; *Nguyen et al., 2016*), identifying neurons across animals is additionally challenging because the nuclear localized markers that are used tend to obscure morphological features that would otherwise assist in neural identification.

An ideal method for finding neural correspondence in *C. elegans* should accommodate two major use cases. The first is tracking neurons within an individual across time as the animal's head moves and deforms. Here, the goal is to be able to say with confidence that a neuron imaged in a volume taken at time $t_1$ is the same as another neuron taken from a volume imaged at time $t_2$. Tracking across time is needed to extract calcium dynamics from neurons during freely moving population calcium imaging (*Venkatachalam et al., 2016*; *Nguyen et al., 2016*; *Lagache et al., 2020*). Additionally, very fast real-time tracking will be needed to bring closed-loop techniques such as brain-machine interfaces (*Clancy et al., 2014*), and optical patch clamping (*Hochbaum et al., 2014*) to moving animals.

**eLife digest** Understanding the intricacies of the brain often requires spotting and tracking specific neurons over time and across different individuals. For instance, scientists may need to precisely monitor the activity of one neuron even as the brain moves and deforms; or they may want to find universal patterns by comparing signals from the same neuron across different individuals.

Both tasks require matching which neuron is which in different images and amongst a constellation of cells. This is theoretically possible in certain 'model' animals where every single neuron is known and carefully mapped out. Still, it remains challenging: neurons move relative to one another as the animal changes posture, and the position of a cell is also slightly different between individuals. Sophisticated computer algorithms are increasingly used to tackle this problem, but they are far too slow to track neural signals as real-time experiments unfold.

To address this issue, Yu et al. designed a new algorithm based on the Transformer, an artificial neural network originally used to spot relationships between words in sentences. To learn relationships between neurons, the algorithm was fed hundreds of thousands of 'semi-synthetic' examples of constellations of neurons. Instead of painfully collated actual experimental data, these datasets were created by a simulator based on a few simple measurements. Testing the new algorithm on the tiny worm *Caenorhabditis elegans* revealed that it was faster and more accurate, finding corresponding neurons in about 10ms.

The work by Yu et al. demonstrates the power of using simulations rather than experimental data to train artificial networks. The resulting algorithm can be used immediately to help study how the brain of *C. elegans* makes decisions or controls movements. Ultimately, this research could allow brain-machine interfaces to be developed.

The second and more general use case is finding neural correspondence across individuals. Often this is to identify the name of a neuron with respect to the connectome (*White et al., 1986*) or a gene expression atlas (*Hammarlund et al., 2018*). Even when a neuron's name cannot be ascertained, being able to identify which neurons are the same across recordings allows researchers to study neural population codes common across individuals.

For both use cases, a method to find neural correspondence is desired that is accurate, fast, requires minimal experimental training data and that generalizes across animal pose, orientation, imaging hardware, and conditions. Furthermore, an ideal method should not only perform well when restricted to neural positioning information but, should also be flexible enough to leverage genetically encoded color labeling information or other features for improved accuracy when available. Multicolor strains are powerful new tools that use multiple genetically encoded fluorescent labels to aid neural identification (*Yemini et al., 2021*; *Toyoshima et al., 2019*) (we use one of those strains, NeuroPAL (*Yemini et al., 2021*), for validating our model). However, some applications, like whole-brain imaging in moving worms, are not yet easily compatible with the multicolor imaging required by these new strains, so there remains a need for improved methods that use position information alone.

A variety of automated methods for *C. elegans* have been developed that address some, but not all these needs. Most methods developed so far focus on finding the extrinsic similarity (*Bronstein, 2007*) between one neuron configuration, called a test, and another neuron configuration called a template. Methods like these deform space to minimize distances between neurons in the template and neurons in the test and then attempt to solve an assignment problem (*Lagache et al., 2018*). For example, a simple implementation would be to use a non-rigid registration model, like Coherent Point Drift (CPD) (*Myronenko and Song, 2010*) to optimize a warping function between neuron positions in the test and template. More recent non-rigid registration algorithms like PR-GLS (*Ma et al., 2016*) also incorporate relative spatial arrangement of the neurons (*Wen et al., 2018*).

Models can also do better by incorporating the statistics of neural variability. NeRVE registration and clustering (*Nguyen et al., 2017*), for example, also uses a non-rigid point set registration algorithm (*Jian and Vemuri, 2011*) to find a warping function that minimizes the difference between a configuration of neurons at one time point and another. But NeRVE further registers the test neurons onto multiple templates to define a feature vector and then finds neural correspondence by

clustering those feature vectors. By using multiple templates, the method implicitly incorporates more information about the range and statistics of that individual animal's poses to improve accuracy.

A related line of work uses generative models to capture the statistics of variability across many individual worms. These generative models specify a joint probability distribution over neural labels and the locations, shapes, sizes, or appearance of neurons identified in the imaging data of multiple individuals (*Bubnis et al., 2019*; *Varol et al., 2020*; *Nejatbakhsh et al., 2020*; *Nejatbakhsh and Varol, 2021*). These approaches are based on assumptions about the likelihood of observing a test neural configuration, given an underlying configuration of labeled neurons. For example, these generative models often begin with a Gaussian distribution over neuron positions in a canonical coordinate system and then assume a distribution over potentially non-rigid transformations of the worm's pose for each test configuration. Then, under these assumptions, the most likely neural correspondence is estimated via approximate Bayesian inference.

The success of generative modeling hinges upon the accuracy of its underlying assumptions, and these are challenging to make for high-dimensional data. An alternative is to take a discriminative modeling approach (*Bishop, 2006*). For example, recent work (*Chaudhary et al., 2021*) has used conditional random fields (CRF) to directly parameterize a conditional distribution over neuron labels, rather than assuming a model for the high-dimensional and complex image data. CRF allows for a wide range of informative features to be incorporated in the model, such as the angles between neurons, or their relative anterior-posterior positions, which are known to be useful for identifying neurons (*Long et al., 2009*). Ultimately, however, it is up to the modeler to select and hand curate a set of features to input into the CRF.

The next logical step is to allow for much richer features to be learned from the data. Artificial neural networks are ideal for tackling this problem, but they require immensely large training sets. Until now, their use for neuron identification has been limited. For example, in one tracking algorithm, artificial neural networks provide only the initialization, or first guess, for non-rigid registration (*Wen et al., 2018*).

Our approach is based on a simple insight: it is straightforward to generate very large semi-synthetic datasets of test and template worms that nonetheless are derived from measurements. We use neural positions extracted from existing imaging datasets, and then apply known, nonlinear transformations to warp those positions into new shapes for other body postures, or other individuals. Furthermore, we simulate the types of noise that appear in real datasets, such as missing or spurious neurons. Using these large-scale semi-synthetic datasets, we train an artificial neural network to map the simulated neural positions back to the ground truth. Given sufficient training data (which we can generate at will), the network learns the most informative features of the neural configurations, rather than requiring the user to specify them by hand.

Importantly, using semi-synthetic data also allows us to train our model even when we completely lack experimentally acquired ground truth data. And indeed, in this work, semi-synthetic data is derived exclusively from measurements that lack any ground truth correspondence either within-, or across animals. All ground truth for training comes only from simulation. Realistic synthetic, semi-synthetic or augmented datasets have been key to cracking other challenging problems in neurosicence (*Parthasarathy et al., 2017*; *Yoon et al., 2017*; *Sun et al., 2018*; *Lee et al., 2020*; *Mathis and Mathis, 2020*; *Pereira et al., 2020*) and have already shown promising potential for tracking neurons (*Wen et al., 2018*).

In this work, we use semi-synthetic data to train a Transformer network, an artificial network architecture that has shown great success in natural language processing tasks (*Vaswani et al., 2017*). Transformers incorporate an attention mechanism that can leverage similarities between pairs of inputs to build a rich representation of the input sequence for downstream tasks like machine translation and sentiment prediction. We reasoned this same architecture would be well-suited to extract spatial relationships between neurons in order to build a representation that facilitates finding correspondence to neurons in a template worm.

Not only is the Transformer well-suited to learning features for the neural correspondence problem, it also obviates the need to straighten (*Peng et al., 2008*) the worm in advance. Until now, existing methods have either required the worm to be straightened in preprocessing (*Bubnis et al., 2019*; *Chaudhary et al., 2021*) or explicitly transformed them during inference (*Varol et al., 2020*;

*Nejatbakhsh et al., 2020*). Straightening the worm is a non-trivial task, and it is especially error-prone for complicated poses such as when the worm rolls along its centerline.

Finally, one of the main advantages of the Transformer architecture is that it permits parallel processing of the neural positions using modern GPU hardware. In contrast to existing methods, which have not been optimized for speed, the Transformer can make real-time predictions once it has been trained. This speed is a necessary step toward bringing real-time applications (*Clancy et al., 2014*; *Hochbaum et al., 2014*) to freely moving animals.

## Results

### Fast deep neural correspondence accurately matches neurons across semi-synthetic individuals

We developed a fast deep neural correspondence (fDNC) model that seeks to find the correspondence between configurations of *C. elegans* neurons in different individuals or in the same individual across time (*Figure 1*). We used a deep learning artificial neural network architecture, called the transformer architecture (*Vaswani et al., 2017*), that specializes at finding pairs of relations in datasets, *Figure 1F*. The transformer architecture identified similarities across spatial relations of neurons in a test and a template to identify correspondences between the neurons.

Within a single individual, neural positions vary as the worm moves, deforms, and changes its orientation and pose. Across isogenic individuals, there is an additional source of variability that arises from the animal's development. In practice, further variability also arises from experimental measurements: neuron positions must first be extracted from fluorescent images (*Figure 1A*), and slight differences in label expression, imaging artifacts, and optical scattering all contribute to errors in segmenting individual neurons.

We created a simulator to model these different sources of variability and used it to generate realistic pairs of empirically derived semi-synthetic animals with known correspondence between their neurons for training our model (*Figure 1B,E*). The simulator took configurations of neuron positions that lacked ground truth from real worms as inputs and then scaled and deformed them, forced them to adopt different poses sampled from real worms, and then introduced additional sources of noise to generate many new semi-synthetic individuals. We then trained our fDNC model on these experimentally derived semi-synthetic individuals of different sizes and poses.

Training our model on the empirically derived semi-synthetic data offered advantages compared to experimentally acquired data. First, it allowed us to train on larger datasets than would otherwise be practical. We trained on $2.304 \times 10^5$ semi-synthetic individuals, but only seeded our simulator with unlabeled neural configurations from experimentally acquired recordings of 12 individuals ($4 \times 10^3$ volumes spread across the 12 individuals, all of which lacked ground-truth correspondence). Second, we did not need to provide ground truth correspondence because the simulator instead generates its own ground truth correspondence between semi-synthetic individuals, thereby avoiding a tedious and error prone manual step. Consequently, no experimentally acquired ground truth correspondence was used to train the model. Later in the work, we use ground truth information from human annotated NeuroPAL (*Yemini et al., 2021*) strains to evaluate the performance of our model, but no NeuroPAL strains were used for training. Importantly, the amount of test data with ground truth correspondence needed for evaluating performance is much smaller than the amount of training data that would be needed for training. Third, by using large and varied semi-synthetic data, we force the model to generalize its learning to a wide range of variabilities in neural positions and we avoid the risks of overtraining on idiosyncrasies specific to our imaging conditions or segmentation. Overall, we reasoned that training with semi-synthetic data should make the model more robust and more accurate across a wider range of conditions, orientations and animal poses than would be practical with experimentally acquired ground-truth datasets.

We trained our fDNC model on $2.304 \times 10^5$ semi-synthetic individuals (*Figure 1C*) and then, after training, evaluated its performance on 2000 additional held-out semi-synthetic pairs of individuals which had not been accessible to the model during training, *Figure 1D* and *Figure 2*. Model performance was evaluated by calculating the accuracy of the models' predicted correspondence with respect to the ground truth in pairs of semi-synthetic individuals. One individual is called the 'test' and the other is the 'template'. Every neuron in the test or template, whichever has fewer is assigned

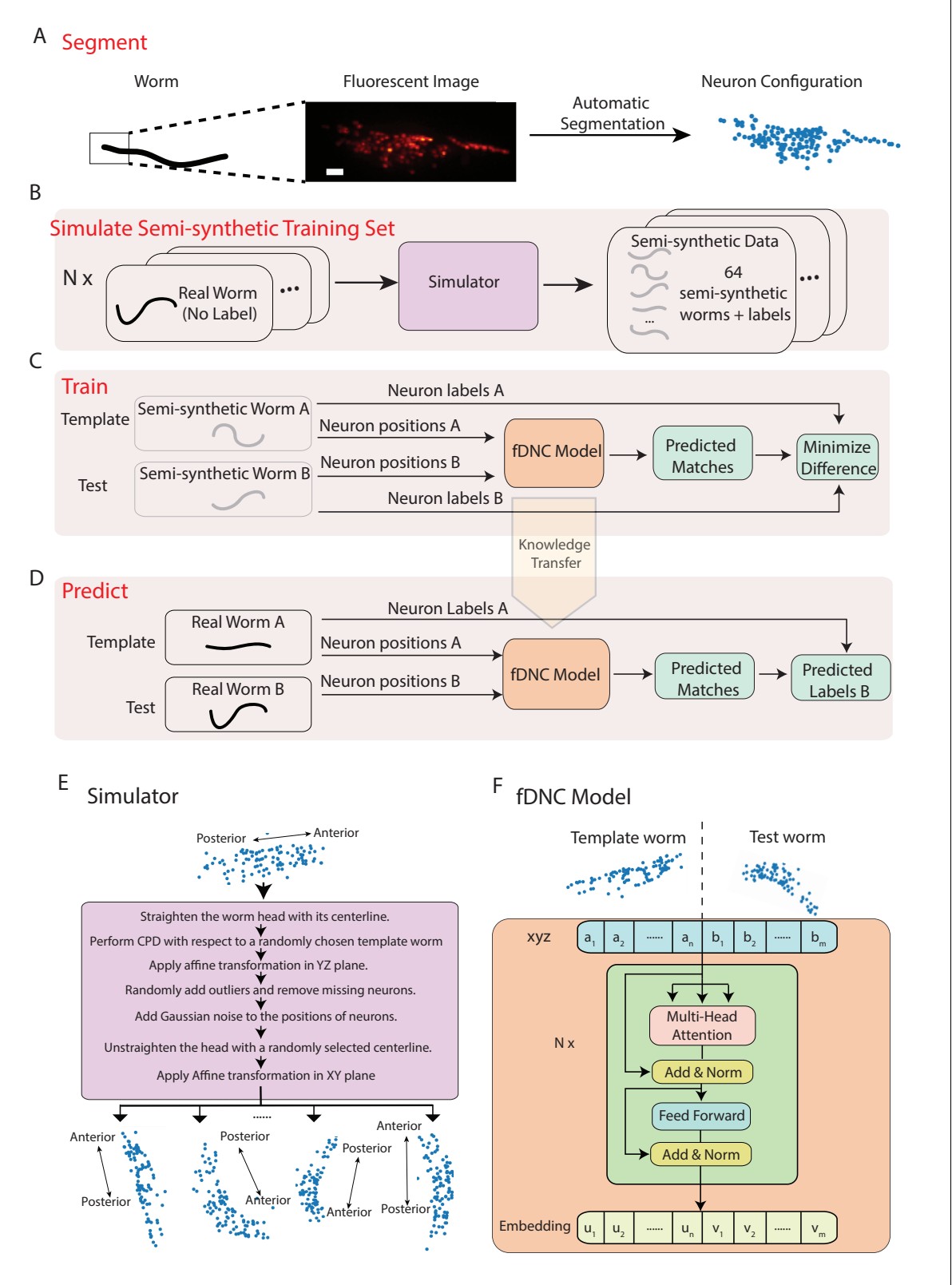

**Figure 1.** Fast deep neural correspondence model. (**A–D**) Schematic of training and analysis pipeline for using the fast Deep Neural Correspondence (fDNC) model to predict correspondence between neurons across individuals. (**A**) Volumetric images of fluorescent labeled neuronal nuclei are segmented to extract neuron positions. (Scale bar, 10 μm). (**B**) Semi-synthetic training data is generated with a simulator. The simulator transforms the neural positions of a real worm and introduces noise to generate new semi-synthetic individuals. Approximately $N = 10^4$ neuron configurations without

*Figure 1 continued on next page*

*Figure 1 continued*

labels from 12 moving worms were used to generate $2.304 \times 10^5$ labeled semi-synthetic worms for training. (**C**) During training, the fDNC model finds optimal internal parameters to minimize the difference between predicted neural correspondence and true correspondence in pairs of semi-synthetic worms. (**D**) Given positions for neurons in real worm A and positions for neurons in real worm B, the trained model predicts correspondences between them. Furthermore,if labels for neurons in A are known, the model can then assign corresponding labels to neurons in worm B. (**E**) Detailed schematic of the simulator from panel B. (**F**) Transformer architecture of the fDNC model. The position features of a template worm with $n$ neurons and a test worm with $m$ neurons are taken as input. The features are computed via a multi-head attention mechanism. 'Add and Norm' refers to an addition and layer normalization step. $a$ and $b$ are neuron positions and $u$ and $v$ are embeddings for the template and test, respectively. We choose the number of layers $N = 6$ and the embedding dimension $d_{emb} = 128$ by evaluating the performance on a held-out validation set.

a match. Accuracy is reported as the number of correctly predicted matches between test and template, divided by the total number of ground truth matches in the test and template pair. Our fDNC model achieved 96.5% mean accuracy on the 2000 pairs of held-out semi-synthetic individuals. We compared this performance to that of Coherent Point Drift (CPD) (*Myronenko and Song, 2010*), a classic registration method used for automatic cell annotation. CPD achieved 31.1% mean accuracy on the same held-out semi-synthetic individuals. Our measurements show that the fDNC model significantly outperforms CPD at finding correspondence in semi-synthetic data. For the rest of the work, we use experimentally acquired human annotated data to evaluate performance.

## fDNC accurately tracks neurons within an individual across time

We next evaluated the fDNC model's performance at tracking neurons within an individual over time, as is needed, for example, to measure calcium activity in moving animals (*Venkatachalam et al., 2016*; *Nguyen et al., 2016*). We evaluated model performance on an experimentally acquired calcium imaging recording of a freely moving *C. elegans* from *Nguyen et al., 2017* in which a team of human experts had manually tracked and annotated neuron positions over time (strain AML32, 1514 volumes, six volumes per second, additional details are describeed in the 'Datasets' section of the 'Materials and methods.'). The recording has sufficiently large animal movement that the average distance a neuron travels between volumes (4.8 μm) is of similar scale to the average distance between nearest neuron neighbors (5.3 μm). The recording was excluded from training and from the set of recordings used by the simulator. We collected neuron configurations from all $n$ time points during this recording to form $n - 1$ pairs of configurations upon which to evaluate the fDNC model. Each pair consisted of a test and template. The template was always from the same time point $t$, while the test was taken to be the volume at any of the other time points. We applied the pre-trained fDNC model to the pairs of neuron configurations and compared the model's predicted correspondence to the ground truth from manual human tracking (*Figure 3*). Across the pairs, the fDNC model showed a mean accuracy of 79.1%. We emphasize that the fDNC model achieved this high accuracy on tracking a real worm using only neuron position information even though it is trained exclusively on semi-synthetic data.

We compared the performance of our fDNC model to that of CPD Registration, and to Neuron Registration Vector Encoding and clustering (NeRVE), a classical computer vision model that we had previously developed specifically for tracking neurons within a moving animal over time (*Nguyen et al., 2017*; *Figure 3C*). fDNC clearly outperformed CPD achieving 79.1% accuracy compared to CPD's 62.7%.

Both CPD and fDNC predict neural correspondence of a test configuration by comparing only to a single template. In contrast, the NeRVE method takes 100 templates, where each one is a different neuron configuration from the same individual, and uses them all to inform its prediction. The additional templates give the NeRVE method extra information about the range of possible neural configurations made by the specific individual whose neurons are being tracked. We therefore compared the fDNC model both to the full NeRVE method and also to a restricted version of the NeRVE method in which NeRVE had access only to the same single template as the CPD or fDNC models. (Under this restriction, the NeRVE method no longer clusters and the method collapses to a series of gaussian mixture model registrations [*Jian and Vemuri, 2011*]). In this way, we could compare the two methods when given the same information. fDNC's mean performance of 79.1% was statistically significantly more accurate than the restricted NeRVE model (mean 73.1%, $p = 1.3 \times 10^{-140}$, Wilcoxon signed rank test). The full NeRVE model that had access to additional

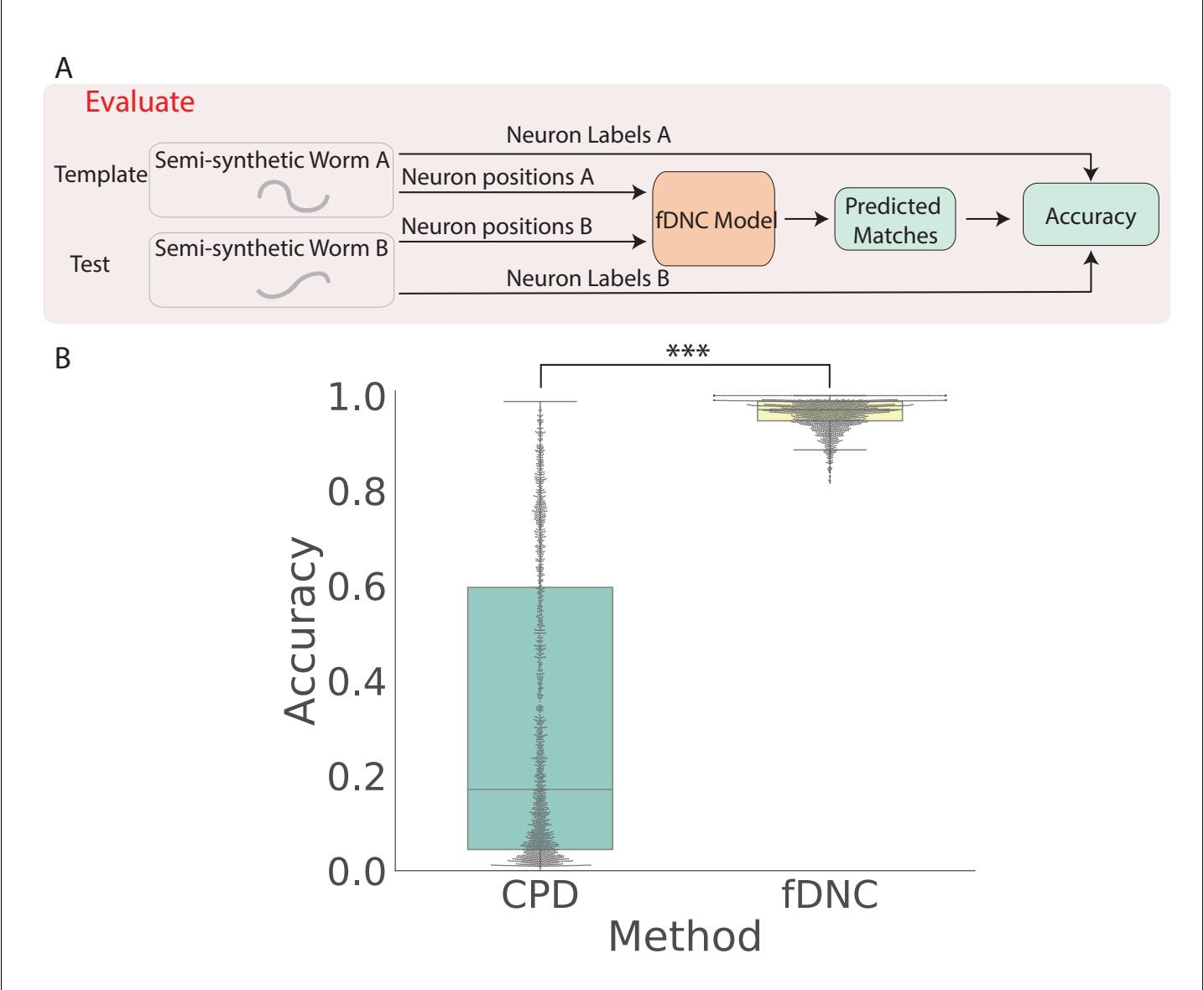

**Figure 2.** fDNC accurately predicts matches between neuron from semi-synthetic worms (**A**) Schematic of evaluation pipeline. fDNC model performance is evaluated on pairs of semi-synthetic worms with known correspondence that had been held out from training. Given neural positions in worms A and B, the model predicts matches between A and B. Accuracy is the number of correctly predicted matches divided by the total number of ground truth matches for the A-B pair. (**B**) Model performance of a Coherent Point Drift Registration (CPD) is compared to the fDNC model on 2000 randomly selected pairs of held-out semi-synthetic individuals, without replacement. ($p = 0$, Wilcoxon signed rank test). The online version of this article includes the following figure supplement(s) for figure 2:

**Figure supplement 1.** fDNC model training curve.

templates outperformed the fDNC model slightly (82.9% $p = 1.5 \times 10^{-102}$, Wilcoxon signed rank test).

Because CPD, NeRVE, and fDNC are all time-independent algorithms, their performance on a given volume is the same, even if nearby volumes are omitted or shuffled in time. One benefit of this approach is that errors from prior volumes do not accumulate over the duration of the recording. To visualize performance over time, we show a volume-by-volume comparison of fDNC's tracking to that of a human (*Figure 3E*). We also characterize model performance on a per neuron basis (*Figure 3D*).

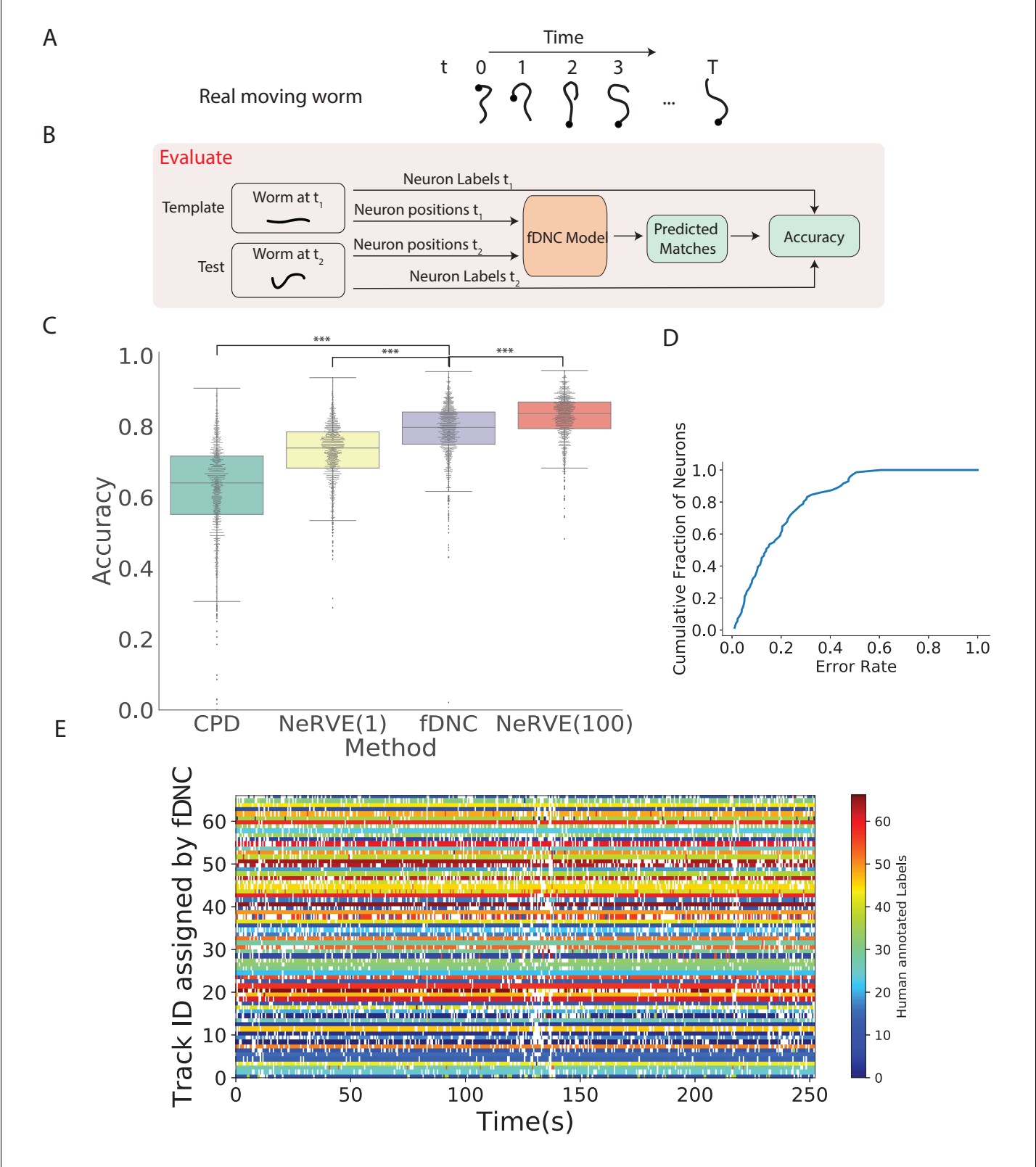

**Figure 3.** Tracking neurons within an individual across time. (**A**) Schematic shows how the pose and orientation of a freely moving animal change with time. Black dot indicates head. (**B**) Pipeline to evaluate the fDNC model at tracking neurons within an individual across time. The fDNC model takes in positional features of a template neuron configuration from one time $t_1$ of a freely moving worm, and predicts the correspondence at another time $t_2$, called the test. Recording is of a moving animal undergoing calcium imaging from ***Nguyen et al., 2017***. Ground truth neuron correspondence are

*Figure 3 continued on next page*

*Figure 3 continued*

provided by manual human annotation. The same time point is used as the template for all 1513 template-test pairs. (**C**) Performance of fDNC and alternative models at tracking neurons within an individual are displayed in order of mean performance. CPD refers to Coherent Point Drift. NeRVE(1) refers to the restricted NeRVE model that has access to only the same template as CPD and fDNC. NeRVE(100) refers to the full NeRVE model which uses 100 templates from the same individual to make a single prediction. A Wilcoxon signed rank significance test of fDNC's performance compared to CPD, NeRVE(1) and NeRVE(100) yields $p = 2.5 \times 10^{-223}, 1.3 \times 10^{-140}$ and $1.5 \times 10^{-102}$, respectively. Boxplots show median and interquartile range. (**D**) fDNC tracking performance by neuron. Cumulative fraction of neurons is shown as a function of the acceptable error rate. (**E**) Detailed comparison of fDNC tracking to human annotation of a moving GCaMP recording from *Nguyen et al., 2017*. Color at each time point indicates the neuron label manually annotated by a human. White gaps indicate that the neuron is missing at that time point. In the case of perfect agreement between human and fDNC, each row will have only a single color or white.

The online version of this article includes the following figure supplement(s) for figure 3:

**Figure supplement 1.** Example of fDNC used to track neurons during free moving calcium imaging.

Finally, we used fDNC to extract whole brain calcium activity from a previously published recording of a moving animal in which two well-characterized neurons AVAL and AVAR were unambiguously labeled with an additional colored fluorophore (*Hallinen et al., 2021*; *Figure 3—figure supplement 1A*, *Video 1*). Calcium activity extracted from neurons AVAL and AVAR exhibited calcium activity transients when the animal underwent prolonged backward locomotion, as expected (*Figure 3—figure supplement 1B*). We conclude that the fDNC model is suitable for tracking neurons across time and performs similarly to the NeRVE method.

In the following sections, we further show that the fDNC method is orders of magnitude faster than NeRVE. Moreover, unlike NeRVE which can only be used within an individual, fDNC is also able to predict the much more challenging neural correspondence across individuals.

## fDNC is fast enough for future real-time applications

Because it relies on an artificial neural network, the fDNC model finds correspondence for a set of neurons faster than traditional methods (*Table 1*). From the time that a configuration of segmented neurons is loaded onto a GPU, it takes only an average of 10 ms for the fDNC model to predict correspondence for all neurons on a 2.4 GHz Intel machine with an NVIDIA Tesla P100 GPU. If not using a GPU, the model predicts correspondence for all neurons in 50 ms. In contrast, on the same hardware it takes CPD 930 ms and it takes NeRVE on average over 10 s. The fDNC model may be a good candidate for potential closed-loop tracking applications because its speed of 100 volumes per second is an order of magnitude faster than the 6–10 volumes per second recording rate typically used in whole-brain imaging of freely moving *C. elegans* (*Nguyen et al., 2016*; *Venkatachalam et al., 2016*). We note that for a complete closed-loop tracking system, fast segmentation algorithms will also be needed in addition to the fast registration and labeling algorithms presented here. The fDNC model is agnostic to the details of the segmentation algorithm so it is well suited to take advantage of fast segmentation algorithms when they are developed.

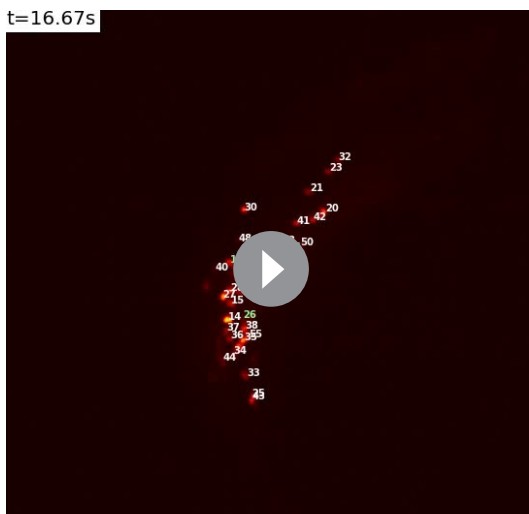

t=16.67s

**Video 1.** Video of neuron tracking during calcium imaging in moving animal. fDNC algorithm is applied to a calcium imaging recording from *Hallinen et al., 2021* (six volumes per second, 200 planes per second). Same recording as in *Figure 3—figure supplement 1*. Images are shown from the RFP channel and show nuclear localized tagRFP in each neuron. For each volume, a single optical plane is shown that contains neuron AVAR (labeled in pink). Labels assigned by fDNC are shown. Color indicates whether the neuron resides in the displayed optical plane (green), or up to two planes above or below (white). The time of the video corresponding to *Figure 3—figure supplement 1* is shown on the left top corner.

https://elifesciences.org/articles/66410#video1

**Table 1.** Time required to predict neural correspondence.

Table shows the measured time per volume required for different models to predict neural correspondence of a single volume. Time required is measured after neuron segmentation is complete and a configuration of neural positions has been loaded into memory. The same hardware is used for all models.

| Method | Time (s/Volume) |
| --- | --- |
| CPD (*Myronenko and Song, 2010*) | 0.93 |
| NeRVE(1) (*Nguyen et al., 2017*) | 10 |
| NeRVE(100) (*Nguyen et al., 2017*) | >10 |
| fDNC [this work] | 0.01 |

The fDNC model uses built-in libraries to parallelize the computations for labeling a single volume, and this contributes to its speed. In particular, each layer of the neural network contains thousands of artificial neurons performing the same computation. Computations for each neuron in a layer can all be performed in parallel and modern GPUs have as many as 3500 CUDA cores.

In practice, the method is even faster for post-processing applications (not-realtime) because it is also parallelizable at the level of each volume. Labeling one volume has no dependencies on any previous volumes and therefore each volume can be processed simultaneously. The number of volumes to be processed in parallel is limited only by the number of volumes that can be loaded onto the memory of a GPU. When tracking during post-processing in this work, we used 32 volumes simultaneously.

## fDNC accurately finds neural correspondence across individuals

Having shown that fDNC performs well at identifying neurons within the same individual, we wanted to assess its capability to identify neurons across different animals, using neural position information alone, as before. Identifying corresponding neurons across individuals is crucial for studying the nervous system. However, finding neural correspondence across individuals is more challenging than within an individual because there is variability in neuronal position from both the animal's movement as well as from development. To evaluate the fDNC model's performance at finding neural correspondence across individuals using only position information, we applied the same semi-synthetically-trained fDNC model to a set of 11 NeuroPAL worms. NeuroPAL worms contain extra color information that allows a human to assign ground truth labels to evaluate the model's performance. Crucially, the fDNC model was blinded to this additional color information. In these experiments, NeuroPAL color information was only used to evaluate performance after the fact, not to find correspondence.

NeuroPAL worms have multicolor neurons labeled with genetically encoded fluorescent proteins (*Yemini et al., 2021*). Only a single volume was recorded for each worm since immobilization is required to capture multicolor information from the NeuroPAL strain. For each of the 11 Neuropal recording, neurons were automatically segmented and manually annotated based on the neuron's position and color features as described in *Yemini et al., 2021* (see *Figure 4A,B*). Across the 11 animals, a human assigned a ground-truth label to a mean of 43% of segmented head neurons, providing approximately 58 labeled neurons per animal (*Figure 4C*, additional details in 'Datasets' section of 'Materials and Methods'). The remaining segmented neurons were not confidently identifiable by the human and thus were left without ground truth labels. We selected as template the recording that contained the largest number of confidently labeled ground turth human annotated neurons. We evaluated our model by comparing its predicted correspondence between neurons in the other 10 test datasets and this template, using only position information (no color information). All 11 ground-truth recordings were held-out in that they were not involved in the generation of the semi-synthetic data that had been used to train the model.

We applied the synthetically trained fDNC model to each pair of held-out NeuroPAL test and template recordings and calculated the accuracy as the number of correctly predicted matches divided by the total number of ground truth matches in the pair. Across the 10 pairs of NeuroPAL

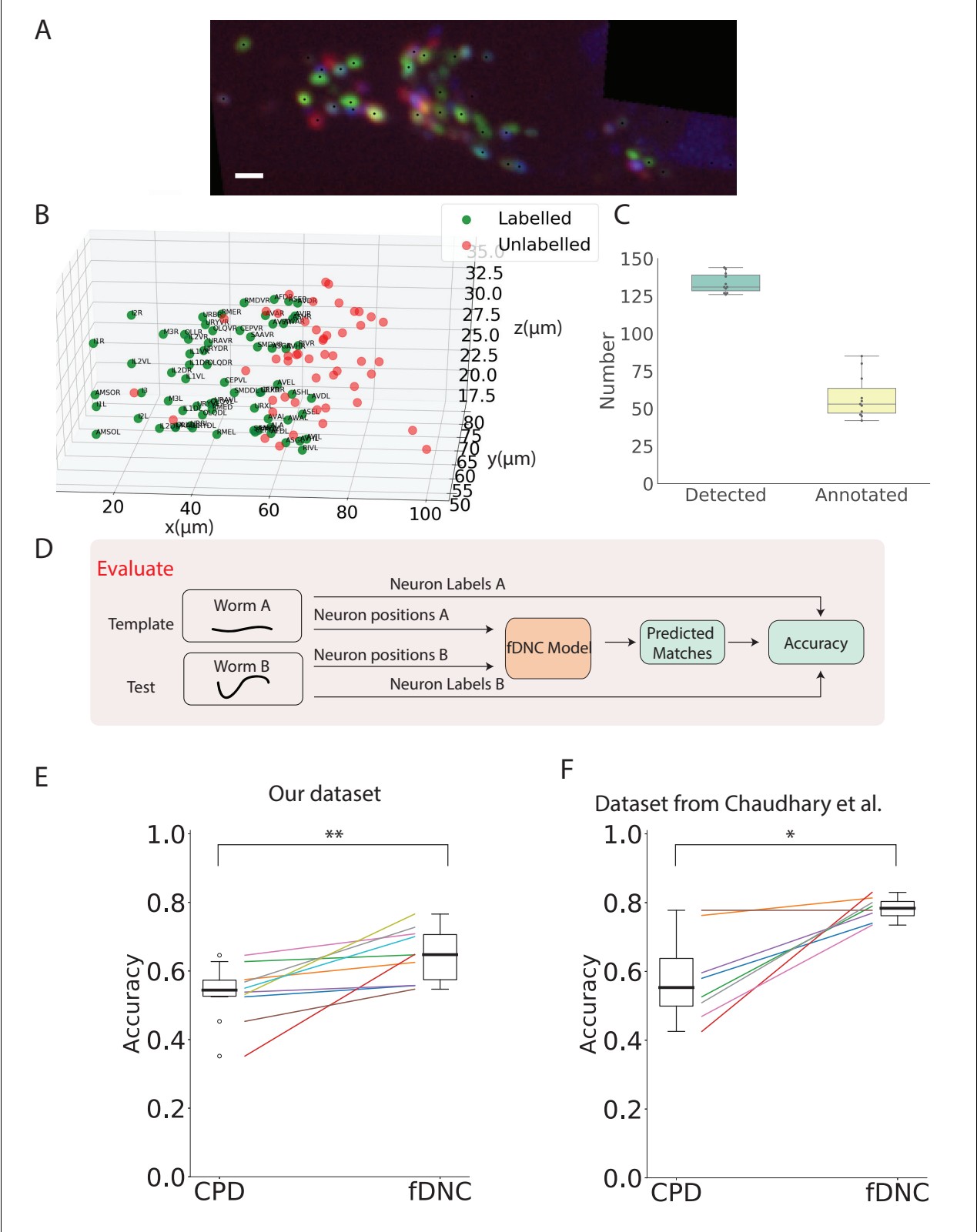

**Figure 4.** fDNC model finds neural correspondence across individuals. (**A**) Fluorescence image shows neuronal nuclei of a NeuroPAL worm. A single optical slice is shown from an optical stack. (Scale bar, 10 μm). Genetically encoded color labels in NeuroPAL animals aid ground truth manual neural identification (*Yemini et al., 2021*) and are used here to evaluate performance. Black dots indicate neurons found via automatic segmentation. (**B**) Locations of all segmented neurons from A. Neurons that additionally have a human annotated label are shown in green. Those that a human was

*Figure 4 continued on next page*

*Figure 4 continued*

unable to label are red. (C) Number of segmented neurons (mean 133.6) and subset of those that were given human annotations (mean 57.5) is shown for 11 NeuroPAL individuals. Box plot shows median and interquartile range. (D) Pipeline to evaluate fDNC model performance across NeuroPAL individual is shown. Predicted labels are compared with human annotated labels to compute accuracy. (E) Performance of the fDNC model and CPD is shown evaluated on NeuroPAL recordings using position information alone. Accuracy is the fraction of labeled neurons present in both test and template that are correctly matched. Performance is evaluated on 10 pairs of 11 recordings, where the template is always the same (Worm A). ($p = 0.005$, Wilcoxon signed-rank test). (F) Performance evaluated on a separate publicly accessible dataset of nine NeuroPAL individuals from *Chaudhary et al., 2021* ($p = 0.018$, Wilcoxon signed-rank test).

recordings using position information alone, the fDNC model had a mean accuracy of 64.1%, significantly higher than the CPD method's accuracy of 53.1% ($p = 0.005$, Wilcoxon signed-rank test).

We wondered whether we could better use the likelihood information about potential matches generated by the algorithm. For each neuron $i$ in the test recording, the fDNC model computes a relative confidence with which that neuron corresponds to each possible neuron $j$ in the template, $p_{ij}$. A Hungarian algorithm finds the most probable match by considering all $p_{ij}$s for all neurons in the test. By default we use this best match in evaluating performance. The $p_{ij}$s also provide the user with a list of alternative matches ranked by the model's estimate of their respective likelihood. We therefore also assessed the accuracy for the top three most likely matches.

Given $i$ and $j$ are ground truth matches, we asked whether the value $p_{ij}$ is among the top three values of the set $p_{ik}$ where $k$ can be chosen from all the neurons in the template. We defined accuracy as the number of instances in which this criteria was met, divided by the number of ground truth matches in the test template pair. When considering the top three neurons, the fDNC model achieves an accuracy of 76.6% using only position information.

## Validating on an alternative dataset

Data quality, selection criteria, human annotation, hardware, segmentation, and preprocessing can all vary from lab to lab making it challenging to directly compare methods. To validate our model against different measurement conditions and to allow for a direct comparison with another recent method, we applied our fDNC model to a previously published dataset of 9 NeuroPAL individuals (*Chaudhary et al., 2021*). This public dataset used different imaging hardware and conditions and was annotated by human experts from a different group. On this public dataset, using position information alone, our method achieved 78.2% accuracy while CPD achieved 58.9%, *Figure 4F*. When assessing the top three candidate accuracy, the fDNC model performance was 91.3%. The fDNC model performance was overall higher on the published dataset than on our newly collected dataset presented here. This suggests that our method performs well when applied to real-world datasets in the literature.

We further sought to compare the fDNC model to the reported accuracy of a recent model called Conditional Random Fields (CRF) from *Chaudhary et al., 2021* by comparing their performance on the same published dataset from that work. There are fundamental differences between the two methods that make a direct comparison of their performance challenging. CRF assigns labels to a test worm. In contrast, fDNC assigns matches between two worms or two configurations, the test and template. To evaluate whether a match is correct using fDNC, we require a ground truth label in both test and template. Consequently, our denominator for accuracy is the intersection of neurons with ground truth labels in test and template. In contrast, the denominator for evaluating accuracy of the CRF model is all neurons with ground truth labels in the test.

When applied to the same dataset in *Chaudhary et al., 2021*, fDNC had an accuracy of 78%. But for the purposes of comparison with CRF this could, in principle, correspond to an accuracy of 61.2–82.5%, depending on how well those neurons in the test that lack ground truth labels in the template were matched. These bounds are calculated for the extreme cases in which neurons with ground truth labels in the test but not in the template are either all matched incorrectly (61.2%) or all matched perfectly (82.5%). Seventy-eight percent is the accuracy under the assumption that those neurons with ground truth labels in the test but not in the template are correctly matched at the same rate as those neurons with ground truth labels in both. In other words, we assume the neurons we have ground truth information about are representative of the ones we don't. For the sake of comparison, we use this assumption to compare fDNC to the published values of CRF (*Table 2*).

**Table 2.** Comparison of across-animal model performance on additional dataset.

Table lists reported mean accuracy of different models evaluated on the same publicly accessible dataset from *Chaudhary et al., 2021*. We note in the text an assumption needed to compare these methods. $N$ indicates the number of template-test pairs used to calculate accuracy. (CRF method uses an atlas as the template, whereas we randomly take one of the nine individuals and designate that as the template). CPD and fDNC performance on this dataset are also shown in *Figure 4F*.

| Method | Accuracy | $N$ | Reported in |
|---|---|---|---|
| CPD | 59% | 8 | This work |
| CRF (open atlas) | ≈40% | 9 | *Chaudhary et al., 2021* |
| CRF (data driven atlas) | 74% | 9 | *Chaudhary et al., 2021* |
| fDNC | 78% | 8 | This work |

fDNC accuracy is higher than the reported performance for the open atlas variant of CRF. Under the specific assumption described above, it is also higher than the data driven atlas variant, although we note that this could change with different assumptions, and we are unable to test for statistical significance. The fDNC method also offers other advantages compared to the CRF approach in that the fDNC method is optimized for speed and avoids the need to transform the worm into a canonical coordinate system. Importantly, compared to the data-driven atlas variant of the CRF, the fDNC model has an advantage in that it does not require assembling a data-driven atlas from representative recordings with known ground-truth labels. Taken together, we conclude that the fDNC model's accuracy is comparable to that of the CRF model while also providing other advantages.

## Incorporating color information

Our method only takes positional information as input to predict neural correspondence. However, when additional features are available, the position-based predictions from the fDNC model can be combined with predictions based on other features to improve overall performance. As demonstrated in *Yemini et al., 2021*, adding color features from a NeuroPAL strain can reduce the ambiguity of predicting neural correspondence. We applied a very simple color model to calculate the similarity of color features between neuron $i$ in the test recording to every possible neuron $j$ in the template. The color model returns matching probabilities, $p_{ij}^{c}$ based on the Kullback-Liebler divergence of the normalized color spectra in a pair of candidate neurons (details described in Materials and methods). The color model is run in parallel to the fDNC model (*Figure 5A*). Overall matching probabilities $p_{ij}^{all}$ that incorporate both color and position information are calculated by combining the color matching probabilities $p_{ij}^{c}$ with the position probabilities $p_{ij}$. The Hungarian algorithm is run on the combined matching algorithm to predict the best matches.

Adding color information increased the fDNC model's average accuracy from 64.1% to 74.7% (*Figure 5B*) when evaluated on our dataset, and improved the accuracy in every recording evaluated. The top three candidate labels attained 92.4% accuracy. Accuracy was calculated from a comparison to human ground truth labeling, as before.

We chose a trivially simple color model in part to demonstrate the flexibility with which the fDNC model framework can integrate information about other features. Since our simple color model utilized no prior knowledge about the distributions of colors in the worm, we would expect a more sophisticated color model, for example, the statistical model used in *Yemini et al., 2021*, to do better. And indeed that model evaluated on a different dataset is reported to have a higher performance with color than our model on our dataset (86% reported accuracy in *Yemini et al., 2021* compared to 75% for the fDNC evaluated here). But that model also performs much worse than fDNC when both are restricted to use only neural position information (50% reported accuracy for *Yemini et al., 2021* compared to 64% for the fDNC). Together, this suggests the fDNC model framework can take advantage of additional feature information like color and still perform relatively well when such information is missing.

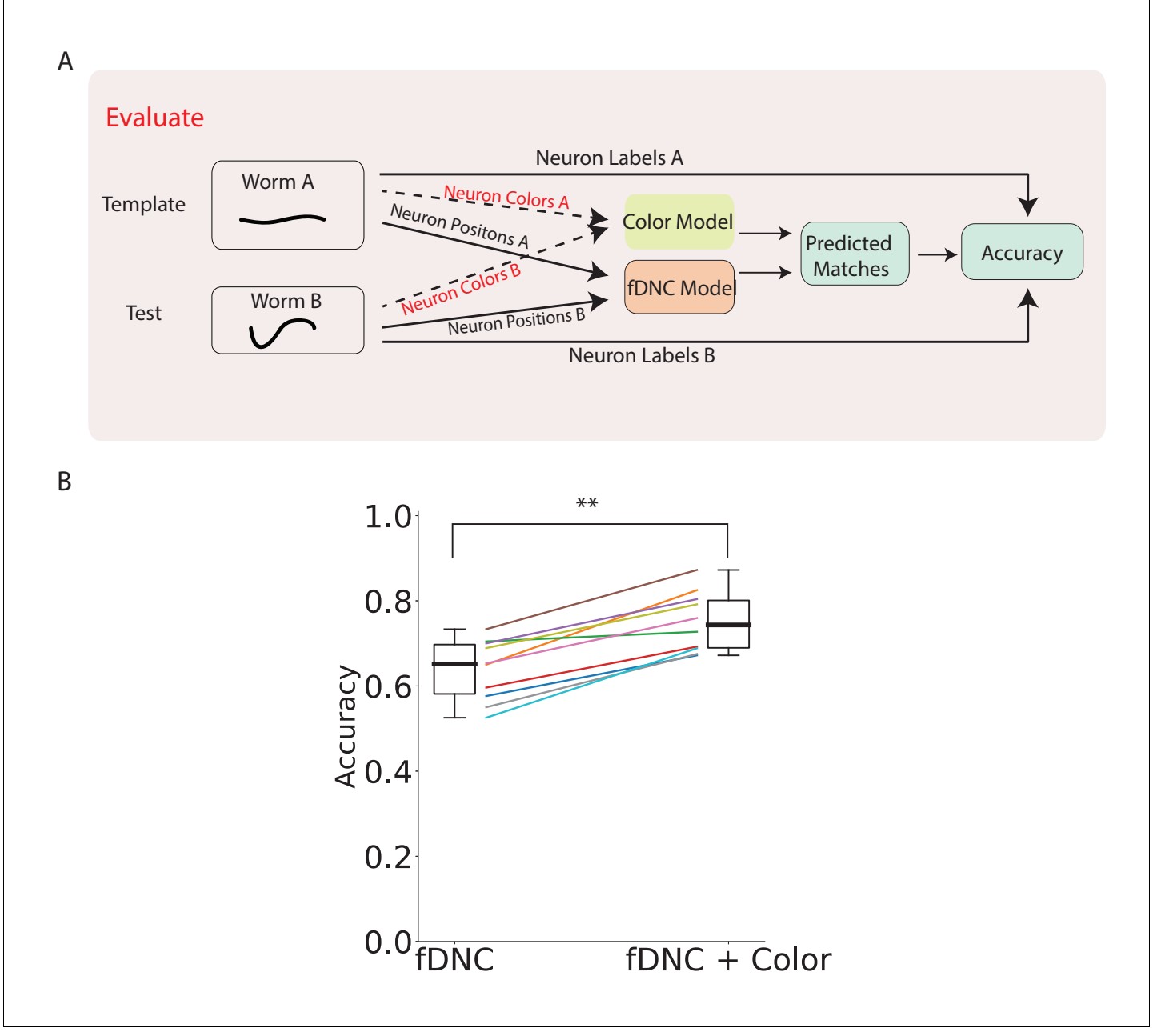

**Figure 5.** fDNC performance when incorporating color features. (**A**) Pipeline to evaluate fDNC performance across animals with additional color features. A simple color model is added in parallel to the fDNC model to use both color and position information from 11 NeuroPAL recordings. Accuracy is calculated from ground truth human annotation and is the fraction of labeled neurons present in both test and template that are correctly matched. Matching probabilities from the color and fDNC models are combined to form the final matching probabilities. (**B**) Accuracy of the position-only fDNC model and the combined fDNC and color model are evaluated on 11 NeuroPAL recordings (same recordings as in *Figure 4*). $p = 5.0 \times 10^{-3}$, Wilcoxon signed rank test.

## Discussion

Identifying correspondence between constellations of neurons is important for resolving two classes of problems: The first is tracking the identities of neurons across time in a moving animal. The second is mapping neurons from one individual animal onto another, and in particular onto a reference atlas, such as one obtained from electron microscopy (*Witvliet et al., 2020*). Mapping onto an atlas

allows recordings of neurons in the laboratory to be related to known connectomic, gene expression, or other measurements in the literature.

The fDNC model finds neural correspondence within and across individuals with an accuracy that is comparable or compares favorable to other methods. The model focuses primarily on identifying neural correspondence using position information alone. For tracking neurons within an individual using only position, fDNC achieves a high accuracy of 79%, while for across individuals using only position it achieves 64% accuracy on our dataset and 78% on a published dataset from another group.

We expect that an upper bound may exist, set by variability introduced during the animal's development, that ultimately limits the accuracy with which any human or algorithm can find correspondence across individuals via only position information. For example, pairs of neurons in one individual that perfectly switch position with respect to another individual will never be unambiguously identified by position alone. It is unclear how close fDNC's performance of 64% on our dataset or 78% on the dataset in *Chaudhary et al., 2021* comes to this hypothetical upper bound, but there is reason to think that at least some room for improvement remains.

Specifically, we do not expect accuracy at tracking within an individual to be fundamentally limited, in part because we do not expect two neurons to perfectly switch position on the timescale of a single recording. Therefore fDNC's 79% accuracy within-individuals suggests room for improving within-individual correspondence, and by extension, across-individual correspondence because the latter necessarily includes all of the variability of the former. One avenue for achieving higher performance could be to improve the simulator's ability to better capture variability of a real testset, for example by using different choices of parameters in the simulator.

Even at the current level of accuracy, the ability to find correspondence across animals using position information alone remains useful. For example, we are interested in studying neural population coding of locomotion in *C. elegans* (*Hallinen et al., 2021*), and neural correspondence at 64% accuracy will allow us to reject null hypotheses about the extent to which neural coding of locomotion is stereotyped across individuals.

The fDNC model framework also makes it easy to integrate other features which further improve accuracy. We demonstrated that color information could be added by integrating the fDNC model with a simple color model to increase overall accuracy. We expect that performance would improve further with a more sophisticated color model that takes into account the statistics of the colors in a NeuroPAL worm (*Yemini et al., 2021*).

The fDNC model framework offers a number of additional advantages beyond accuracy. First, it is versatile and general. The same pre-trained model performed well at both tracking neurons within a freely moving individual across time and at finding neural correspondence across different individuals. Without any additional training, it achieved even higher accuracy on a publicly accessible dataset acquired on different hardware with different imaging conditions from a different group. This suggests that the framework should be applicable to many real-world datasets. The model provides probability estimates of all possible matches for each neuron. This allows an experimenter to consider a collection of possible matches such as the top three most likely.

In contrast to previous methods, an advantage of the fDNC method is that it does not require the worm to be straightened, axis aligned, or otherwise transformed into a canonical coordinate system. This eliminates an error-prone and often manual step. Instead, the fDNC model finds neural correspondence directly from neural position information even in worms that are in different poses or orientations.

Importantly, the model is trained entirely on semi-synthetic data, which avoids the need for large experimentally acquired ground truth datasets to train the artificial neural network. Acquiring ground truth neural correspondence in *C. elegans* is time consuming, error prone, and often requires manual hand annotation. The ability to train the fDNC model with semi-synthetic data derived from measurements alleviates this bottleneck and makes the model attractive for use with other organisms with stereotyped nervous systems where ground truth datasets are similarly challenging to acquire.

The model is also fast and finds neural correspondence of a new neural configuration in 10 ms. The development of fast algorithms for tracking neurons are an important step for bringing real-time closed loop applications such as optical brain-machine interfaces (*Clancy et al., 2014*) and optical patch clamping (*Hochbaum et al., 2014*) to whole-brain imaging in freely moving animals. By

contrast, existing real-time methods for *C. elegans* in moving animals are restricted to small subsets of neurons, are limited to two-dimensions, and work only at low spatial resolution (*Leifer et al., 2011*; *Stirman et al., 2011*; *Kocabas et al., 2012*; *Shipley et al., 2014*). We note that to be used in a real-time closed loop application, our fDNC model would need to be combined with faster segmentation algorithms because current segmentation algorithms are too slow for real-time use. Because segmentation can be easily paralellized, we expect that faster segmentation algorithms will be developed soon.

Many of the advantages listed here stem from the fDNC model's use of the transformer architecture (*Vaswani et al., 2017*) in combination with supervised learning. The transformer architecture, with its origins in natural language processing, is well suited to find spatial relationships within a configuration of neurons. By using supervised learning on empirically derived semi-synthetic training data of animals in a variety of different poses and orientations, the model is forced to learn relative spatial features within the neurons that are informative for finding neural correspondence across many postures and conditions. Finally, the transformer architecture leverages recent advances in GPU parallel processing for speed and efficiency, which is an important step toward future real-time applications.

# Materials and methods

## Key resources table

| Reagent type (species) or resource | Designation | Source or reference | Identifiers | Additional information |
|---|---|---|---|---|
| Strain, strain background (*C. elegans*) | AML320 | this work | | See *Table 4* |
| Strain, strain background (*C. elegans*) | OH15262 | *Yemini et al., 2021* | RRID:WB-STRAIN:WBStrain00047397 | |

## Datasets

### Recordings used by simulator to generate semi-synthetic data

Our model was trained on a semi-synthetic training dataset that was simulated from 4000 volumes spread across recordings of 12 freely moving animals of strain AML32 acquired during calcium imaging. The recordings fed to the simulator had no ground truth correspondence either within or across animals. Each recording had originally contained approximately 3000 volumes recorded at six volumes/s. The recordings fed to the simulator were set aside after use by the simulator and were never re-used for evaluating model performance.

### Datasets used to evaluate performance

The model's performance was evaluated on various types of datasets with ground truth correspondence, as shown in *Table 3*. All these recordings were held-out in the sense that they were never used for training. Some of these recordings had ground truth correspondence within an individual over time, while others had ground truth correspondence across individuals. One of the NeuroPAL (*Yemini et al., 2021*) datasets is a published dataset from an independent research group (*Chaudhary et al., 2021*). One of the calcium imaging datasets, from *Hallinen et al., 2021*, had no ground truth correspondence and served to demonstrate the model's ability to extract calcium activity.

## Data availability

Neural configurations acquired as part of this study have been posted in an Open Science Foundation repository with DOI:10.17605/OSF.IO/T7DZU available at https://dx.doi.org/10.17605/OSF.IO/T7DZU. The publicly accessible dataset from *Chaudhary et al., 2021* is available at https://github.com/shiveshc/CRF_Cell_ID, commit 74fb2feeb50afb4b840e8ec1b8ee7b7aaa77a426. Datasets from *Nguyen et al., 2017* and *Hallinen et al., 2021* are publicly available in repositories associated with their respective publications.

**Table 3.** Ground truth content, by dataset.

Table lists ground truth properties for each dataset used in this work to evaluate the model. None of the datasets listed here were used for training. 'Vol' refers to volume and 'indiv' refers to individuals. Ground truth 'matches pair$^{-1}$' indicates the average number of ground truth matches for random pairs of test and template, which is a property of the ground truth dataset, and does not depend on the model tested.

| | Held-out semi-synthetic testset | Ca$^{2+}$ imaging | NeuroPAL | NeuroPAL | Ca$^{2+}$ imaging |
|---|---|---|---|---|---|
| Figure | *Figure 2* | *Figure 3* | *Figures 4* and *5* | *Figure 4F* | *Figure 3—figure supplement 1* |
| Type | - | moving | immobile | immobile | moving |
| Correspondence | across indiv | within indiv | across indiv | across indiv | within indiv |
| Ground Truth | simulator | human | human | human | - |
| Ground truth matches pair$^{-1}$ | 85.7 | 64.4 | 50.1 | 50.5 | - |
| Ground truth labels vol$^{-1}$ | 102.1 | 69.2 | 57.5 | 64.3 | - |
| Segmented neurons vol$^{-1}$ | 114.1 | 118.4 | 133.6 | 118.8 | 131.1 |
| Total Vols | 2000 | 1514 | 11 | 9 | 1400 |
| Individuals | 2000 | 1 | 11 | 9 | 1 |
| Vols indiv$^{-1}$ | 1 | 1514 | 1 | 1 | 1400 |
| Vols s$^{-1}$ | - | 6 | - | - | 6 |
| Strain | - | AML32 | AML320 (via OH15262) | OH15495 | AML310 |
| Reference | this work | *Nguyen et al., 2017* | this work | *Chaudhary et al., 2021* | *Hallinen et al., 2021* |

## Strains

Those strains used to create new datasets presented in this work are listed in Key Resources. All strains mentioned in this study, including those involved in previously published datasets, are listed in *Table 4*. All strains express a nuclear localized red fluorescent protein in all neurons. All but strains OH15495 and OH15262 also express nuclear localized GCaMP6s in all neurons. NeuroPAL (*Yemini et al., 2021*) strains further express many additional fluorophores.

## Imaging

To image neurons in the head of freely moving worms, we used a dual-objective spinning-disk based tracking system (*Nguyen et al., 2016*) (Yokogawa CSU-X1 mounted on a Nikon Eclipse TE2000-S). Fluorescent images of the head of a worm were recorded through a 40x objective with both 488- and 561 nm excitation laser light as the animal crawled. The 40x objective translated up and down along the imaging axis to acquire 3D image stacks at a rate of 6 head volumes/s.

To image neurons in the immobile multi-color NeuroPAL worms (*Yemini et al., 2021*), we modified our setup by adding emission filters in a motorized filter wheel (Prior ProScan-II), and adding a Stanford Research Systems SR474 shutter controller (with SR475 shutters) to programmatically illuminate the worm with different wavelength laser light. We use three lasers of different wavelengths: 405 nm (Coherent OBIS-LX 405 nm 100 mW), 488 nm (Coherent SAPPHIRE 488 nm 200 mW), and 561 nm (Coherent SAPPHIRE 561 nm 200 mW). Only one laser at a time reached the sample, through a 40x oil-immersion objective (1.3 NA, Nikon S Fluor). The powers measured at the sample, after spinning disk and objective, were 0.14 mW (405 nm), 0.35 mW (488 nm), and 0.36 mW (561 nm). In the spinning disk unit, a dichroic mirror (Chroma ZT405/488/561tpc) separated the excitation from the emission light. The latter was relayed to a cooled sCMOS camera (Hamamatsu ORCA-Flash 4.0 C11440-22CU), passing through the filters mounted on the filter wheel (*Table 5*). Fluorescent images were acquired in different 'channels', that is, different combinations of excitation wavelength, emission filter, and camera exposure time (*Table 6*). The acquisition was performed using a

**Table 4.** List of all strains mentioned in this work.

| Strain | RRID | Genotype | Notes | Ref |
|---|---|---|---|---|
| AML32 | RRID:WB-STRAIN: WBStrain00000192 | wtfIs5[Prab-3::NLS::GCaMP6s; Prab-3::NLS::tagRFP] | | *Nguyen et al., 2017* |
| AML310 | RRID:WB-STRAIN: WBStrain00048356 | wtfIs5[Prab-3::NLS::GCaMP6s; Prab-3::NLS::tagRFP]; wtfEx258 [Prig-3::tagBFP::unc-54] | | *Hallinen et al., 2021* |
| AML320 | | (otIs669[NeuroPAL] V 14x; wtfIs145 [pBX + rab-3::his-24:: GCaMP6::unc-54]) | derived from OH15262 | this work |
| OH15262 | RRID:WB-STRAIN: WBStrain00047397 | otIs669[NeuroPAL] | | *Yemini et al., 2021* |
| OH15495 | RRID:WB-STRAIN: WBStrain00047403 | otIs696[NeuroPAL] | | *Yemini et al., 2021*; *Chaudhary et al., 2021* |

custom software written in LabVIEW that specifies the sequence of channels to be imaged, and controls shutters, filter wheel, piezo translator, and camera. After setting the z position, the software acquires a sequence of images in the specified channels.

## Preprocessing and segmentation

We extracted the position of individual neurons from 3D fluorescent images to generate a 3D point cloud (*Figure 1A*). This process is called segmentation and the fDNC model is agnostic to the specific choice of the segmentation algorithm. Segmentation was always performed on tagRFP, never on GCaMP.

For recordings of strains AML32, we used a segmentation algorithm adopted from *Nguyen et al., 2017*. We first applied a threshold to find pixels where the intensities are significantly larger than the background. Then, we computed the 3D Hessian matrix and its eigenvalues of the intensity image. Candidate neurons were regions where the maximal eigenvalue was negative. Next, we searched for the local intensity peaks in the region and spatially disambiguated peaks in the same region with a watershed separation based on pixel intensity.

For recordings of NeuroPAL strains, we used the same segmentation algorithm as in *Yemini et al., 2021*. The publicly accessible dataset from *Chaudhary et al., 2021* used in *Figure 4* had already been segmented prior to our use.

## Generating semi-synthetic point clouds with correspondence for training

We developed a simulator to generate a large training set of semi-synthetic animals with known neural correspondence. The simulator takes as its input the point clouds collected from approximately 4000 volumes spread across 12 recordings of freely moving animals. Each recording contains roughly 3000 volumes. For each volume, the simulator performs a series of stochastic deformations and transformations to generate 64 new semi-synthetic individuals where the ground truth correspondence between neurons in the individuals and the original point cloud is known. A total of $2.304 \times 10^5$ semi-synthetic point clouds were used for training.

The simulator introduces a variety of different sources of variability and real-world deformations to create each semi-synthetic point cloud (*Figure 1B,E*). The simulator starts by straightening the worm in the XY plane using its centerline so that it now lies in a canonical worm coordinate system. Before straightening, Z is along the optical axis and XY are defined to be perpendicular to the optical axis and are arbitrarily set by the orientation of the camera. After straightening, the animal's

**Table 5.** List of emission filters for multicolor imaging.

| Filter label | Filters (Semrock part n.) |
|---|---|
| F1 | FF01-440/40 |
| F2 | FF01-607/36 |
| F3 | FF02-675/67 + FF01-692/LP |

**Table 6.** Imaging channels used.

| Channel | Excitation λ (nm) | Emission window (nm) [filter] | Primary fluorophore |
|---------|-------------------|-------------------------------|---------------------|
| ch0 | 405 | 420–460 [F1] | mtagBFP |
| ch1 | 488 | 589–625 [F2] | CyOFP |
| ch2 | 561 | 589–625 [F2] | tagRFP-t |
| ch3 | 561 | 692–708 [F3] | mNeptune |

posterior-anterior axis lies along the X axis. To introduce animal-to-animal variability in relative neural position, a non-rigid transformation is applied to the neuron point cloud against a template randomly selected from recordings of the real observed worms using coherent point drift (CPD) (*Myronenko and Song, 2010*). To add variability associated with rotation and distortion of the worm's head in the transverse plane, we apply a random affine transformation to the transverse plane. To simulate missing neurons and segmentation errors, spurious neurons are randomly added, and some true neurons are randomly removed, for up to 20% of the observed neurons. To introduce variability associated with animal pose, we randomly deform the centerline of the head. Lastly, to account for variability in animals' size and orientation, a random affine transformation in the XY plane is applied that rescaled the animal's size by up to 5%. With those steps, the simulator deforms a sampled worm and generates a new semi-synthetic worm with different orientation and posture while maintaining known correspondence.

Centerlines generated by the simulator were directly sampled from recordings of real individuals. The magnitude of added Gaussian noise was arbitrarily set to have a standard deviation of 0.42 µm.

## Deep neural correspondence model

### Overview and input

The deep neural correspondence model (fDNC) is an artificial neural network based on the Transformer (*Vaswani et al., 2017*) architecture (*Figure 1C*) and is implemented in the automatic differentiation framework PyTorch (*Paszke et al., 2017*). The fDNC model takes as input the positional coordinates of a pair of worms, a template worm $a$, and test worm, $b$ (*Figure 1F*). For each worm, approximately 120 neurons are segmented and passed to the fDNC model. The input neuron sequences are randomly shuffled for both template worm and test worm. This eliminates the possibility that the information from the original sequence order is used.

### Architecture

The model works as an encoder, which maps the input neuron coordinates $(a_1, a_2, ..., a_n, b_1, b_2, ..., b_m)$ to continuous embeddings $(u_1, u_2, ...u_n, v_1, v_2, ..., v_m)$. The model is composed of a stack of $N = 6$ identical layers. Each layer consists of two sub-layers: a multi-head self-attention mechanism (*Vaswani et al., 2017*) and a fully connected feed-forward network. The multi-head attention mechanism is the defining feature of the transformer architecture and makes the architecture well-suited for finding relations in sequences of data, such as words in a sentence or, in our case, spatial locations of neurons in a worm. Each head contains a one-to-one mapping between the nodes in the artificial network and the *C. elegans* neurons. In the transformer architecture, features of a previous layer are mapped via a linear layer into three attributes of each node, called the query, the key and the value pairs. These attributes of each node contain high dimensional feature vectors which, in our context, represent information about the neuron's relative position. The multi-head attention mechanism computes a weight for each pair of nodes (corresponding to each pair of *C. elegans* neurons). The weights are calculated by performing a set computation on the query and key. The output is calculated by multiplying this resultant weight by the value. In our implementation, we set the number of heads in the multi-head attention module to be eight and we set the dimension of our feature vectors to be 128. We chose the best set of the hyperparameters (details in Training section) by evaluating on a validation set, which is distinct from the training set and also from any data used for evaluation. A residual connection (*He et al., 2016*) and layer normalization (*Jl et al., 2016*) are employed for each sub-layer, as is widely used in artificial neural networks.

## Calculating probabilities for potential matches

The fDNC model generates a high dimensional ($d = 128$) embedding $u_i$ for neuron $i$ from the template worm and $v_j$ for the neuron $j$ from the test worm. The similarity of a pair of embeddings, as measured by the inner product $\langle u_i, v_j \rangle$, determines the probability that the pair is a match. Specifically, we define the probability that neuron $i$ in the template worm matches neuron $j$ in the test worm as $p_{ij}$, where

$$p_{ij} = \frac{e^{\langle u_i, v_j \rangle}}{\sum_{k=1}^{m} e^{\langle u_i, v_k \rangle}}. \tag{1}$$

Equivalently, the vector $p_i = (p_{i1}, \ldots, p_{im})$ is modeled as the 'softmax' function of the inner products between the embedding of neuron $i$ and the embeddings of all candidate neurons $1, \ldots, m$. The softmax output is non-negative and sums to one so that $p_i$ can be interpreted as a discrete probability distribution over assignments of neuron $i$.

We also find the most probable correspondence between the two sets of neurons by solving a maximum weight bipartite matching problem where the weights are given by the inner products between test and template worm embeddings. This is a classic combinatorial optimization problem, and it can be solved in polynomial time using the Hungarian algorithm (**Kuhn, 1955**).

## End-user output

The fDNC model returns two sets of outputs to the end user. One is the algorithm's estimate of the most probable matches for each neuron in the test worm; that is, the solution to the maximum weight bipartite matching problem described above. The other is an ordered list of alternative candidate matches for each individual neuron in the test worm and their probabilities ranked from most to least probable.

## Training

The model was trained on $2.304 \times 10^5$ semi-synthetic animals derived from recordings of 12 individuals. The model was trained only once and the same trained model was used throughout this work.

Training is as follows. We performed supervised learning with ground truth matches provided by the semi-synthetically generated data. A cross-entropy loss function was used. If neuron $i$ and neuron $j$ were matched by human, the cross-entropy loss function favors the model to output $p_{ij} = 1$. If neuron $i$ and neuron $j$ were not matched, the loss function favors the model to output $p_{ij} = 0$. The model was trained for 12 hr on a 2.40 GHz Intel machine with NVIDIA Tesla P100 GPU.

We trained different models with different hyperparameters and chose the one with best performance. The training curve for each model we trained is shown in *Figure 2—figure supplement 1*. All the models converged after 12 hr of training. We show the performance of trained models on a held-out validation set consisting of 12,800 semi-synthetic worms in *Table 7*. We chose the model with 6 layers and 128 dimensional embedding space since it reaches the highest performance and increasing the complexity of the model did not appear to increase the performance dramatically.

**Table 7.** Model validation for hyperparameters selection.

Table lists losses of models with different hyperparameter values. $N$ represents the number of layers for the transformer architecture. $d_{emb}$ is the dimension of the embedding space. The loss shown is the average cross entropy loss evaluated on a held out validation set.

| Accuracy $d_{emb}$ N | 32 | 64 | 128 |
|---|---|---|---|
| 4 | 83.1% | 88.4% | 90.7% |
| 6 | 86.3% | 94.6% | 96.8% |
| 8 | 90.5% | 96.4% | 96.8% |

## Evaluating model performance and comparing against other models

To evaluate performance, putative matches are found between template and test, and compared to ground truth. Every segmented neuron in the test or template (whichever has fewer) is assigned a match. Accuracy is defined as the number of proposed matches that agree with ground truth, divided by the total number of ground truth matches. The number of ground truth matches is a property of the dataset used to evaluate our model, and is listed in *Table 3*.

### Coherent Point Drift

We use Coherent Point Drift (CPD) Registration (*Myronenko and Song, 2010*) as a baseline with which to compare our model's performance. In our implementation, CPD is used to find the optimal non-rigid transformation to align the test worm with respect to the template worm. We then calculated the distance for each pair of the neurons from the transformed test worm and the template worm. We used the Hungarian algorithm (*Kuhn, 1955*) to find the optimal correspondence that minimizes the total squared distance for all matches.

## Color model

The recently developed NeuroPAL strain (*Yemini et al., 2021*) expresses four different genetically encoded fluorescent proteins in specific expression patterns to better identify neurons across animals. Manual human annotation based on these expression patterns serves as ground truth in evaluating our model's performance at finding across-animal correspondence. In *Figure 5B*, we also explored combining color information with our fDNC model. To do so, we developed a simple color matching model that operated in parallel to our position-based fDNC model. Outputs of both models were then combined to predict the final correspondence between neurons.

Our color matching model consists of two steps: First, the intensity of each of the color channels is normalized by the total intensity. Then the similarity of color for each pair of neurons is measured as the inverse of the Kullback–Leibler divergence between their normalized color features.

To calculate the final combined matching matrix, we add the color similarity matrix to the position matching log probability matrix from our fDNC model. The similarity matrix of color is multiplied by a factor $\lambda$. We chose $\lambda = 60$ so that the amplitude of values in the similarity matrix of color is comparable to our fDNC output. We note the matching results are not particularly sensitive to the choice of $\lambda$. The most probable matches are obtained by applying Hungarian algorithm on the combined matching matrix.

## Code

Source code in Python is provided for the model, for the simulator, and for training and evaluation. A jupyter notebook with a simple example is also provided. Code is available at https://github.com/XinweiYu/fDNC_Neuron_ID (*Yu, 2021*; copy archived at swh:1:rev:19c678781cd11a17866af7b6348ac0096a168c06).

## Acknowledgements

We thank Eviatar Yemini and Oliver Hobert of Columbia University for strain OH15262. We acknowledge productive discussions with John Murray of University of Pennsylvania. This work used computing resources from the Princeton Institute for Computational Science and Engineering. Research reported in this work was supported by the Simons Foundation under awards SCGB #543003 to AML and SCGB #697092 to SWL; by the National Science Foundation, through an NSF CAREER Award to AML (IOS-1845137) and through the Center for the Physics of Biological Function (PHY-1734030); by the National Institute of Neurological Disorders and Stroke of the National Institutes of Health under award numbers R21NS101629 to AML and R01NS113119 to SWL; and by the Swartz Foundation through the Swartz Fellowship for Theoretical Neuroscience to FR. Some strains are being distributed by the CGC, which is funded by NIH Office of Research Infrastructure Programs (P40 OD010440).

## Additional information

### Funding

| Funder | Grant reference number | Author |
|---|---|---|
| Simons Foundation | 543003 | Andrew M Leifer |
| Simons Foundation | 697092 | Scott W Linderman |
| National Science Foundation | IOS-184537 | Andrew M Leifer |
| National Science Foundation | PHY-1734030 | Andrew M Leifer |
| National Institutes of Health | R21NS101629 | Andrew M Leifer |
| National Institutes of Health | 1R01NS113119 | Scott W Linderman |
| National Institutes of Health | P40 OD010440 | Matthew S Creamer |
| Swartz Foundation | Swartz Fellowship for Theoretical Neuroscience | Francesco Randi |

The funders had no role in study design, data collection and interpretation, or the decision to submit the work for publication.

### Author contributions

Xinwei Yu, Conceptualization, Software, Formal analysis, Validation, Investigation, Visualization, Methodology, Writing - original draft, Writing - review and editing; Matthew S Creamer, Investigation, Writing - review and editing, Collected data; Francesco Randi, Resources, Designed optics and related software libraries; Anuj K Sharma, Resources, Writing - review and editing, Performed all transgenics; Scott W Linderman, Conceptualization, Funding acquisition, Writing - review and editing; Andrew M Leifer, Conceptualization, Supervision, Funding acquisition, Project administration, Writing - review and editing

### Author ORCIDs

Xinwei Yu (iD) https://orcid.org/0000-0002-8699-3546
Matthew S Creamer (iD) https://orcid.org/0000-0002-9458-0629
Francesco Randi (iD) https://orcid.org/0000-0002-6200-7254
Anuj K Sharma (iD) https://orcid.org/0000-0001-5061-9731
Scott W Linderman (iD) http://orcid.org/0000-0002-3878-9073
Andrew M Leifer (iD) https://orcid.org/0000-0002-5362-5093

### Decision letter and Author response

Decision letter https://doi.org/10.7554/eLife.66410.sa1
Author response https://doi.org/10.7554/eLife.66410.sa2

## Additional files

### Supplementary files

• Transparent reporting form

### Data availability

All datasets generated as part of this work have been deposited in a public Open Science Foundation repository DOI: https://doi.org/10.17605/OSF.IO/T7DZU.

The following dataset was generated:

| Author(s) | Year | Dataset title | Dataset URL | Database and Identifier |
|---|---|---|---|---|
| Yu X, Cramer M, Randi F, Sharma A, Linderman S, Leifer | 2021 | fDLC_Neuron_ID_C.elegans | http://dx.doi.org/10.17605/OSF.IO/T7DZU | Open Science Framework, 10.17605/OSF.IO/T7DZU |

The following previously published dataset was used:

| Author(s) | Year | Dataset title | Dataset URL | Database and Identifier |
|---|---|---|---|---|
| Nguyen JP, Linder AN, Plummer GS, Shaevitz JW, Leifer AM | 2017 | Tracking Neurons in a Moving and Deforming Brain Dataset | https://dx.doi.org/10.21227/H2901H | IEEE DataPorts, 10.21227/H2901H |

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

snp:e5e96d9309b26413616ec249ec37d7abdf7e9f97;anchor=swh:1:rev:
19c678781cd11a17866af7b6348ac0096a168c06

