## [Decision Letter]

**Acceptance summary:**

This manuscript will be of interest to *C. elegans* neuroscientists and also biologists interested in methodological innovations in live imaging. The method described in the paper is clever and elegant, and the solution to the neuron correspondence problem is significant because it is another step toward closed-loop neural perturbation experiments in mobile worms.

**Decision letter after peer review:**

Thank you for submitting your article "Fast deep learning correspondence for neuron tracking and identification in *C. elegans* using synthetic training" for consideration by *eLife*. Your article has been reviewed by 3 peer reviewers, including Gordon J Berman as the Reviewing Editor and Reviewer #1, and the evaluation has been overseen by Ronald Calabrese as the Senior Editor.

Essential revisions:

1) Many details about the training and characterization are missing. There should be more supplemental info accompanying the manuscript on training the network, characterizations of robustness against noise and errors, verifications – including quantifications like scrambling the data sequence, etc. Not having the information creates a big uncertainty on how to evaluate how close the work is to being usable in a real scenario (i.e., one where segmentation is also available). For whole-brain imaging, there are so many perturbations that can throw off tracking algorithms – segmentation error, cells sometimes "show up" and sometimes "disappear" (the newer version of GCaMPs are really low in baseline), etc. Not seeing these explored systematically concerned the reviews as they didn't have a good sense of whether they have attempted to look at these issues. Thus, we ask the authors to add additional figures and quantifications along these lines to buttress the manuscript's claims.

2) There also was a paucity of details on the network training itself. The methods section mentions, in passing, that some hyper-parameter choices were made on a validation set. Which hyper-parameters were selected in this way, and what ranges of parameters were tried? In this exploration, did the authors observe that network performance was sensitive to some hyper-parameters?

3) Since speed was a crucial criterion for model design, was there any trade-off between network size and running speed, such that future hardware may possibly achieve higher accuracy without further conceptual advances? Similarly, the results all report the performance of one trained network, which the authors report taking half a day to train. This is both a long time and not so long. Were other networks trained and not described, or did they fail to converge? If so, how often do these networks train successfully if using different initial conditions, and do their performances differ?

4) Relatedly, the reviewers also thought that the clarity of the algorithm performance is lacking. For instance, there are no training curves shown for the algorithm. Adding details on the network performance/training into supplementary materials would be beneficial.

5) The majority of the paper uses the authors' own data, which has unique features and structures, leading the reviewers to wonder if the presented results are as generalizable as the authors claim. For example, the authors only accounted for cells that are present in all data sets. What would the numbers look like if they divided by all cells that might show up in any of the animals (a significantly larger number)? The authors do bring up the issue of coverage and accuracy trade-off, but did not really explore this issue at all – the reviewers thought that this point is critical to whole-brain imaging, as the data are rather noisy. So this will have to be addressed using other whole-brain data (maybe published data from another lab, e.g. the Zimmer lab), re-do analysis on the neuron identification part, and more characterization on the coverage-accuracy trade-off. If using the Zimmer data, for example, they should be able to show that they can get the same temporal PCs. If the algorithm is very generalizable and extremely fast and quite accurate as the authors claim, it should be fairly simple to use it on a real whole-brain experiment data set and show that meaningful conclusions can come of it. Without this, one should not make such claims that "The method is fast and predicts correspondence in 10 ms making it suitable for future real-time applications."

6) Related to this point, the reviewers thought that the tracking having an ~80% accuracy is not a meaningful goal. First, this accuracy is an average, and it has no bearing on whether a cell can be *continuously* tracked. The traces may be broken, and worse, wrong cells are linked together. This 80% does not guarantee anything at this point. Having an accuracy on a per-frame basis is not the main goal, and there are existing data from the authors themselves and others where this point could be validated. One would have to show that the traces are similar, and better yet, the temporal PCs are similar. The tracking having 80% accuracy cannot be used for optogenetics at all. It is not meaningful to fire the laser at cells with 20% uncertainty in their identities and carry out any meaningful experiments. Thus, either additional validation on the continuity of the tracked accuracy needs to be provided, or the text on optogenetics needs to be significantly toned down or removed.

7) What is the practical use-case for this cross-individual correspondence, since 65.8% means there are still a lot of errors. Perhaps the authors can discuss (even with some back-of-the-envelope estimates) how much this means for an experiment that compares neural activity between two different worms? What is an experiment that may require doing this fast correspondence estimation between two worms in real-time? Practically speaking, how often would one need to compute correspondences between pairs of frames between two worms? Would the overall correspondence be better if more volumes from each animal were used to find a consensus, or would the authors recommend using NeRVE in that case?

8) "Recording" was used multiple times in the text and it's not clear whether they are time series or single volumes. For instance, it is not clear what exactly are the "12 individual animals" used for generating the training data. Are they single time frames or are they video? If videos, how many frames? It is not clear the NeuroPAL data sets are videos or single volumes.

9) Relatedly, if many time points of 12 individual animals are used to generate training data, this is not fully synthetic. The basis of the training data from many worm heads holds a lot of information. The question is also whether all (any) of the augmentation components are necessary or useful. There should be a full characterization of the differential benefit of the different augmentations from not augmenting at all. Calling it synthetic data (e.g., line 566) may be somewhat of a misnomer.

*Reviewer #1:*

In this submission, the authors introduce a new methodology for tracking neural correspondences in calcium imaging of freely moving *C. elegans* using the transformer architecture. The method presents produces state-of-the-art assignment accuracies in a manner that is significantly faster and more robust than existing approaches, potentially allowing for real-time tracking applications once other aspects of the computer vision pipeline become faster. The authors demonstrate the ability of their method on data within and between individuals (using the NeuroPALworm lines), as well as on synthetic control data.

*Reviewer #2:*

Yu et al. developed a deep neural network model with the goal of solving two challenging problems in live imaging of neural activity in mobile *C. elegans*. They call their method fast Deep Learning Correspondence (fDLC), and it simultaneously achieves (1) identification of the same neurons within one animal in a movie, and (2) identification of corresponding neurons between two animals. These problems are difficult because worms change poses as they move, including wiggles within a horizontal plane as well as rolls, and there may be developmental variability between individual worms. Many past approaches have relied on computationally `straightening` the worm into a canonical coordinate system, or on generative models that rely on manually curated features. Instead, fDLC takes a neural network approach; the model builds on the transformer architecture, popularized by its success in natural language process (NLP). To circumvent the prohibitively large quantities of data typically required to train such models, the authors used a relatively modest experimental dataset and synthetically augmented the training data, simulating worm-like movements and imaging conditions to generate arbitrarily large training sets. They then tested the trained model on a variety of data not used on the training, reporting good accuracy. The fDLC approach described here is particularly impressive because of its speed -- it computes correspondances among ~100 neurons in 10 msec per volume on relatively standard GPU hardware.

Strengths

The paper is generally clearly written, the methods and results are well presented, and the figures are concise summaries of the results. The introduction gives a thorough and thoughtful review of the related literature and how this work relates to previous methods. I particularly appreciate the authors have made their code publicly available. I believe the paper describes a valuable contribution that will be of significant impact in the study of *C. elegans* neuroscience, as it solves a series of related technical challenges whose solution will open the door for more bold experiments.

The method described is well suited for the problem, and the performance described is impressive when compared to the closest methods available in the literature. The results are all well demonstrated and justify the conclusions.

Weaknesses

The strengths of the fDLC method are to enable real-time neural perturbations and to allow direct comparisons between different worms. However, as the authors point out, the real-time experiments are currently still intractable because cell segmentation remains slow. Further, direct comparisons between different worms remain to be demonstrated as an application of fDLC.

On the first application, the true impact of fDLC may have to wait for further development of real-time cell segmentation. This seems like an imminently achievable technology. On the second application, it remains to be shown whether the 65.8% accuracy -- while quite impressive -- is sufficient to allow novel analyses and insights to be gained. For instance, if one were analyzing a dataset of 10 separately imaged worms, the overall accuracy of identifying an individual corresponding neuron among these 10 animals may be significantly lower.

– Perhaps I'm being a bit nit-picky on terminology, but the use of the phrase `transfer learning` in the abstract (also in Figure 1) seems a bit of a stretch. Am I interpreting correctly that the `transfer` is between the train and test sets, without any further refinement? In what way is this `transfer learning` beyond the standard machine learning use of the test/train split?

– The methods sections mentions in passing, that some hyper-parameter choices were made on a validation set. Which hyper-parameters were selected in this way, and what ranges of parameters were tried? In this exploration, did the authors observe that network performance was sensitive to some hyper-parameters?

– In the tracking results of the same worm across time, the fDLC approach treats each set of coordinates as independent measurements and does not explicitly use any temporal information. Nevertheless, I would imagine that segmented neuron positions from adjacent frames of the same movie, when the worm has not moved its pose by much, may be easier to track than pairs of frames picked at random. Is this true? What about frames that are 2, 3, etc. samples apart?

– The acronym `fDLC` may be easily confused with some modification of DeepLabCut. While this work is also a deep learning based tracking software, I think mistaking this method for DeepLabCut may be not desirable.

*Reviewer #3:*

This manuscript describes a deep learning model for tracking neurons in *C. elegans* worms; a side utility of the algorithm is described to be for neuron identification. The problems it is trying to address are significant as there is a need for fast neuron tracking in moving *C. elegans* whole brain imaging; the premise of the work of using synthetic data for training is interesting. The manuscript has several significant deficiencies, including claims not fully supported by evidence and overreaching conclusions.

Major strengths:

1. The idea of using augmentation to real data to generate training sets for ML model is interesting, particularly in situations where data are hard to come by.

2. fDLC's speed is attractive for the use cases.

Major weaknesses:

1. For tracking to have ~80% accuracy is not meaningful. First, this accuracy is an average, and it has no bearing on whether a cell can be *continuously* tracked. The traces may be broken, and worse, wrong cells are linked together. This 80% does not guarantee anything at this point. Having an accuracy on per-frame basis is not useful at all. To actually have an impact on tracking, traces have to be shown, and these traces need to be verified. There are existing data from the authors themselves and others. One would have to show that the traces are similar, and better yet, the temporal PCs are similar. The tracking having 80% accuracy cannot be used for optogenetics at all. It is not meaningful to fire the laser at cells with 20% uncertainty in their identities and carry out any meaningful experiments. This claim does not make sense. The text on optogenetics needs to be significantly toned down, or better yet, removed.

2. What the Transformer network learned with the data is unclear. The paper does not show exactly what the Transformer network has learned – what features of the data are important? This is critical, as it is possible that another form of information is actually being learned from the data. For instance, in training where the hand-curated cells are used, the cells may be entered in a particular order. It is therefore possible that the Transformer network is learning the order of which the cells are entered, rather than the actual spatial relationships. To show that the Transformer network is really learning something meaningful in the data, one would have to scramble the order of the data and show that the results are not different.

3. The authors stressed that the learning does not require users to prescribe what to look for, but the warping, transformation, noises added are in essence adding information in user-defined way. This claim does not make sense. In the text, the authors also use language such as "roughly matched (their) estimate of variability observed by eye". This is not rigorous and seems dangerous. Exact details and rationales of choices for the warping, transformation, noises added, etc need to be included and fully justified.

4. Clarity of algorithm performance is lacking. For instance, there are no training curves shown for the algorithm.

5. Related, importantly, the accuracy of the algorithm must be very much data-dependent. Sources that can perturb a perfect scenario need to be examined. For instance, how would cells' activities in GCaMP recordings affect accuracy? How would segmentation error affect accuracy? It is not possible to evaluate the real-world utility if these issues are not explored. For all we know, it could be the best data that are fed to the algorithm that is used to calculate the accuracies here.

6. The authors stated that there is a trade-off between accuracy and coverage. This is an important point, but the authors did not fully characterize such trade-off (related to the accuracy comment above); nor was the coverage assumption/definition that went into each part of the work clearly stated. In the tracking part, what would the coverage be? How is it defined? Comparisons to literature algorithm for neuron identification is should not be done when the coverage is also not well defined, i.e. the denominators for the percentages in table 4 are ill-defined.

7. Clarity of the experimental data is lacking. "Recording" was used multiple times in the text and it's not clear whether they are time series or single volumes. For instance, it is not clear what exactly are the "12 individual animals" used for generating the training data. Are they single time frames or are they video? If videos, how many frames? It is not clear the NeuroPAL data sets are videos or single volumes.

8. Related to the issue above, if many time points of 12 individual animals are used to generate training data, this is not at all synthetic. The basis of the training data from many worm heads holds a lot of information. The question is also whether all (any) of the augmentation components are necessary or useful. There should be a full characterization of the differential benefit of the different augmentations from not augmenting at all. Calling it synthetic data in my opinion is a misnomer (e.g. line 566).

9. Generally speaking, if the algorithm is very generalizable and extremely fast, and quite accurate as the authors claim, it should be fairly simple to use it on a real whole-brain experiment data set and show that meaningful conclusions can come of it. Without this, one should not make such claims that "The method is fast and predicts correspondence in 10 ms making it suitable for future real-time applications."

[Editors' note: further revisions were suggested prior to acceptance, as described below.]

Thank you for resubmitting your work entitled "Fast deep neural correspondence for tracking and identifying neurons in *C. elegans* using semi-synthetic training" for further consideration by *eLife*. Your revised article has been evaluated by Ronald Calabrese (Senior Editor) and a Reviewing Editor.

The manuscript has been improved but there are some remaining issues that need to be addressed, as outlined below:

Essential Revisions:

1. The data are from the authors themselves, not peer-reviewed, and not independently validated. The authors did not use, for instance, the Zimmer lab's data; the reason why was unclear to the reviewers. Also, the authors themselves have at least one volume of sensible data from their own previous work (NeRVE, Nguyen et al., 2017) in which they actually performed PCA on the GCaMP data. Applying fDNC to that set of data and showing that PCAs are comparable would make their claim much stronger.

2. The accuracy is a central claim in the paper. It is good that the authors now define what accuracy is in the text, but it is still confusing. A match between the template and the test does not assign a name necessarily -- unless the template neurons already have labels/identities from the ground truth information. From the text, it seems that the template is used as the reference with identities already assigned and that only the neurons common to both the test and template are considered since the denominator of the accuracy is defined as "the total number of ground truth matches". (Another interpretation of the definition would suggest that neurons that both the template and the test got wrong but matches each other would have been counted as accurate?!)

There are two issues – the definition is not applicable for some other methods and that this definition is artificially favorable for fDNC.

a. In Table 2, the authors compare the accuracies of fDNC to that of CPD and CRF (ref 3). This is not appropriate. fDNC and CPD both use template matching, while CRF does not. This is to say that the accuracy definition is not the same for these methods.

b. The accuracy of fDNC is artificially more favorable. NeuroPAL datasets do not reliably identify the same neurons. When using one NeuroPAL dataset as template, and another as the test set, the matches are on the order of 70-80%. The definition of accuracy the authors use, therefore, is artificially high (by some significant percentage). The errors associated in neurons not common to the test and the template are discounted.

c. The coverage and the accuracy discussion should be restored.

3. Implying that fDNC is not "data-privileged" is false (page 13). fDNC is not naive – information from 4000 volumes from 12 animals is there, and fDNC must use a known annotated NeuroPAL dataset as a template, and therefore there is information again (e.g. variability of positions etc). Revising the discussion around this point is important.

*Reviewer #1:*

I thank the authors for their thoughtful revisions and especially for providing additional methodological details and caveats. I think that it would make a good addition to the literature.

*Reviewer #2:*

I thank the authors for their careful and detailed responses to comments and concerns. I especially appreciate the additional methodological details on the training of their model and the clarified definition of how performance is evaluated. I think this work is an interesting and valuable contribution to the literature, and another substantial step in achieving real-time manipulations in this popular experimental organism.

*Reviewer #3:*

The revised manuscript is improved for many of the details, including the data used and how the model was constructed, which are good for reproducibility.

The responses are unsatisfying in a few major places:

1. One of the central concerns from the previous round of review is on whether the algorithm performs well enough for tracking. The revision is unsatisfactory.

a. The authors were asked to apply their tracking to real data to show that the tracking results can generate meaningful data. The authors misunderstood the request as asking for biological insights. The intention is to VALIDATE, not to generate new insights. In fact, that's precisely the reason to apply the algorithm/model to well-curated data that are already peer-reviewed and published.

b. Figure 3 and the supplemental data were a step in the right direction, but are still unsatisfactory. Figure 3E now shows the tracking accuracy; as expected, the errors are sporadic. Some neurons appear to be ok while others not. This is THE reason PCA was asked in the last round. Figure 3 supplement showing AVAL/R traces are not enough to demonstrate the traces are sensible. It is anecdotal. AVA are among the most "obvious" neurons; peaks correlating to reversal behavior made the cells easy to identify and the tracking errors very easily ignored. The question is whether the rest of the neurons (>99% of them) can give sensible traces.

c. The data are from the authors themselves, not peer reviewed and not independently validated. The authors dodged the request to use, for instance, the Zimmer lab's data; the reason is unclear to me. Also, if anything, the authors themselves have at least one volume of sensible data from their own previous work (NeRVE, Nguyen et al., 2017) in which they actually did PCA on the GCaMP data. The least they can do is to apply fDNC to that set of data and show that PCAs are comparable.

d. Tracking is tracking, and should not be confounded with the discussion on neuron identification. The accuracy for tracking purposes should be discussed separately.

2. The authors cited a biorxiv paper (ref 22) and glossed over its contribution. This paper is now published (https://elifesciences.org/articles/59187). The major contribution of 3DeeCellTracker is also a deep learning algorithm for tracking cells, and the paper also dealt with the sort of data this manuscript addresses. There is no discussion and no comparison.

a. In fact, Wen et al. directly compared 3DeeCellTracker performance with other algorithms, including even on the dataset from Nguyen et al. (2017). The accuracies reported in Wen et al., are quite favorable (>90%). This data set should be directly compared.

b. fDNC may be faster, which suggests a trade-off between speed and accuracy. It seems pertinent to include this discussion.

3. The accuracy is a central claim in the paper. It is good that the authors now define what accuracy is in the text, but it is still confusing. A match between the template and the test does not assign a name necessarily, UNLESS the template neurons already have labels/identities from the ground truth information. From the text, it seems that the template IS used as the reference with identities already assigned, and that only the neurons common to both the test and template are considered since the denominator of the accuracy is defined as "the total number of ground truth matches". (Another interpretation of the definition would suggest that neurons that both the template and the test got wrong but matches each other would have been counted as accurate?!)

There are two issues – the definition is not applicable for some other methods and that this definition is artificially favorable for fDNC.

a. In Table 2, the authors compare the accuracies of fDNC to that of CPD and CRF (ref 3). This is not appropriate. fDNC and CPD both use template matching, while CRF does not. This is to say that the accuracy definition is not the same for these methods.

b. The accuracy of fDNC is artificially more favorable. NeuroPAL datasets do not reliably identify the same neurons. When using one NeuroPAL dataset as template, and another as the test set, the matches is on the order of 70-80%. The definition of accuracy the authors use, therefore, is artificially high (by some significant percentage). The errors associated in neurons NOT common to the test and the template are discounted.

c. It seems that the coverage and the accuracy discussion should be restored.

---

## [Author Response]

Essential revisions:1) Many details about the training and characterization are missing. There should be more supplemental info accompanying the manuscript on training the network, characterizations of robustness against noise and errors, verifications – including quantifications like scrambling the data sequence, etc. Not having the information creates a big uncertainty on how to evaluate how close the work is to being usable in a real scenario (i.e., one where segmentation is also available). For whole-brain imaging, there are so many perturbations that can throw off tracking algorithms – segmentation error, cells sometimes "show up" and sometimes "disappear" (the newer version of GCaMPs are really low in baseline), etc. Not seeing these explored systematically concerned the reviews as they didn't have a good sense of whether they have attempted to look at these issues. Thus, we ask the authors to add additional figures and quantifications along these lines to buttress the manuscript's claims.

Thank you for these suggestions:

– To characterize training we have added Figure 2 – Supplementary Figure 1 showing training curves and Table 7 showing final performance for all hyperparameters that we explored.

– We had added the following new figures and a video to further demonstrate performance on real-world calcium imaging datasets of moving *C. elegans*:

– Figure 3E shows a volume-by-volume comparison of the model’s assigned neural identities to those that were manually annotated in a freely moving calcium imaging dataset.

– Figure 3D shows a breakdown of model performance by neuron for that dataset.

– Figure 3 – Supplementary Figure 1 shows calcium activity extracted from a recently published recording of whole-brain activity of a moving worm from (Hallinen et al., 2021). In that recording an additional fluorophore unambiguously labels AVAL and AVAR. We show that AVAL and AVAR’s calcium activity show expected transients.

– Video 1 shows labeled neurons over time from the same dataset in Figure 3 – Supplementary Figure 1.

– Regarding concerns related to GCaMP’s baseline activity: We note that all animals in this study expressed both RFP and GCaMP. Segmentation is performed only on RFP, thus we do not anticipate GCaMP activity to have an impact. We now clarify this in the text: “Segmentation was always performed on tagRFP, never on GCaMP.”

– Regarding the scrambling of data sequences: all of the input neuron sequences used in this work have been randomly shuffled both for the training set and the test set, thereby preventing the model from learning any information from original sequence order. We now clarify this in the text. “The input neuron sequences have been randomly shuffled for both template worm and test worm. This eliminates the possibility that the information from the original sequence order is used.”

We also note that, by design, the transformer model can’t extract information from the order of input sequence without additional position embeddings due to its permutation invariance property.

– Regarding challenging our model with realistic noise: For semi-synthetic worms, up to 20% of the total neurons are randomly added or abandoned. As mentioned above, we have also now added an additional real-world recording and show that calcium activity of a well characterized neuron pair, AVAL and AVAR exhibit expected transients.

2) There also was a paucity of details on the network training itself. The methods section mentions, in passing, that some hyper-parameter choices were made on a validation set. Which hyper-parameters were selected in this way, and what ranges of parameters were tried? In this exploration, did the authors observe that network performance was sensitive to some hyper-parameters?

In Table 7, we now report the hyperparameters that we tried. The hyper-parameters include the dimensionality of hidden space (32, 64, 128) and the number of layers (4, 6, 8) for the transformer architecture. All the models we tried converged ( see new Figure 2 – Supplementary Figure 1). We have now added a paragraph of text: “We trained different models with different hyperparameters and chose the one with best performance. The training curve for each model we trained is shown in Figure 2 Supplementary Figure 1. All the models converged after 12 hours of training. We show the performance of trained models on a held-out validation set consisting of 12,800 semi-synthetic worms in Table 7. We chose the model with 6 layers and 128 dimensional embedding space since it reaches the highest performance and increasing the complexity of the model did not appear to increase the performance dramatically. “

3) Since speed was a crucial criterion for model design, was there any trade-off between network size and running speed, such that future hardware may possibly achieve higher accuracy without further conceptual advances? Similarly, the results all report the performance of one trained network, which the authors report taking half a day to train. This is both a long time and not so long. Were other networks trained and not described, or did they fail to converge? If so, how often do these networks train successfully if using different initial conditions, and do their performances differ?

The model already achieves a very high accuracy over our semi-synthetic data ( 96.5%). This suggests that the current bottleneck for improving performance on real data likely has less to do with speed and more to do with the semi-synthetic data’s ability to capture the full variability of real measurements. We now mention this in the discussion:

“Therefore fDNC's 79% accuracy within-individuals suggests room for improving within-individual correspondence, and by extension, across-individual correspondence because the latter necessarily includes all of the variability of the former. One avenue for achieving higher performance could be to improve the simulator's ability to better capture variability of a real dataset, for example by using different choices of parameters in the simulator. ”

All the models with different hyperparameters converged. We have added text to describe convergence and convergence time.

“We trained different models with different hyperparameters and chose the one with best performance. The training curve for each model we trained is shown in Figure 2 Supplementary Figure 1. All the models converged after 12 hours of training. We show the performance of trained models on a held-out validation set consisting of 12,800 semi-synthetic worms in Table 7. We chose the model with 6 layers and 128 dimensional embedding space since it reaches the highest performance and increasing the complexity of the model did not appear to increase the performance dramatically.”

4) Relatedly, the reviewers also thought that the clarity of the algorithm performance is lacking. For instance, there are no training curves shown for the algorithm. Adding details on the network performance/training into supplementary materials would be beneficial.

Training curves have been added, see Figure 2 – Supplementary Figure 1.

5) The majority of the paper uses the authors' own data, which has unique features and structures, leading the reviewers to wonder if the presented results are as generalizable as the authors claim. For example, the authors only accounted for cells that are present in all data sets. What would the numbers look like if they divided by all cells that might show up in any of the animals (a significantly larger number)? The authors do bring up the issue of coverage and accuracy trade-off, but did not really explore this issue at all – the reviewers thought that this point is critical to whole-brain imaging, as the data are rather noisy. So this will have to be addressed using other whole-brain data (maybe published data from another lab, e.g. the Zimmer lab), re-do analysis on the neuron identification part, and more characterization on the coverage-accuracy trade-off. If using the Zimmer data, for example, they should be able to show that they can get the same temporal PCs. If the algorithm is very generalizable and extremely fast and quite accurate as the authors claim, it should be fairly simple to use it on a real whole-brain experiment data set and show that meaningful conclusions can come of it. Without this, one should not make such claims that "The method is fast and predicts correspondence in 10 ms making it suitable for future real-time applications."

We demonstrate generalizability by showing that the model performs well on all recordings in a published dataset from another group (Chaudhary et al., 2021) and on a previously published GCaMP dataset (Figure 3). We have now added an additional calcium imaging dataset, Figure 3 – Supplementary Figure 1 and corresponding video, Video 1. Extracted calcium dynamics of neurons AVAL and AVAR from this dataset exhibit expected transients.

We thank the reviewers for pointing out that our definition of accuracy may have unnecessarily caused confusion. We have made changes that should remove ambiguity.

– We now use a more straightforward definition of accuracy consistently across the entire manuscript. And we note that we do account for all neurons: “To evaluate performance, putative matches are found between template and test, and compared to ground truth. Every segmented neuron in the test or template (whichever has fewer) is assigned a match. Accuracy is defined as the number of proposed matches that agree with ground truth, divided by the total number of ground truth matches.”

– In Table 3, we now list the average number of ground truth matches for pairs of test and templates sampled from each dataset:

“The number of ground truth matches is a property of the dataset used to evaluate our model, and is listed in Table 3.”

– We regenerated all figures using this definition of accuracy. Numerical values are all now slightly different (e.g. in the worst case 65.8% became 64.1%), but our conclusions remain the same.

– We removed all discussion of what we previously had termed “coverage” because it no longer applies to this definition of accuracy.

We support our claim that the method is “fast and predicts correspondence in 10 ms” by providing evidence of the model’s speed (Table 1) and accuracy (Figure 3).

We disagree with the comment that we “should not make such claims” without additional “meaningful conclusions.” We have followed *eLife*’s author guidelines for Tools and Resources submissions: “Tools and Resources articles do not have to report major new biological insights or mechanisms, but it must be clear that they will enable such advances to take place, for example, through exploratory or proof-of-concept experiments.” Our experiments demonstrate the potential of this method for new discovery, and we are excited to use this method in all of our future scientific investigations.

6) Related to this point, the reviewers thought that the tracking having an ~80% accuracy is not a meaningful goal. First, this accuracy is an average, and it has no bearing on whether a cell can be *continuously* tracked. The traces may be broken, and worse, wrong cells are linked together. This 80% does not guarantee anything at this point. Having an accuracy on a per-frame basis is not the main goal, and there are existing data from the authors themselves and others where this point could be validated. One would have to show that the traces are similar, and better yet, the temporal PCs are similar. The tracking having 80% accuracy cannot be used for optogenetics at all. It is not meaningful to fire the laser at cells with 20% uncertainty in their identities and carry out any meaningful experiments. Thus, either additional validation on the continuity of the tracked accuracy needs to be provided, or the text on optogenetics needs to be significantly toned down or removed.

– Figure 3E now demonstrates the extent to which neurons are tracked continuously.

– Figure 3D now shows a breakdown of accuracy per-neuron.

– To further demonstrate that the model works with real-world data, in Figure 3 Supplementary Figure 1 we now apply the method to an additional previously published real-world recording of a moving animal during calcium imaging recording and show that well-characterized neurons AVAL and AVAR exhibit expected calcium transients.

– We have added text to note that the model tracks neurons without regard to time- or history-dependence: “Because CPD, NeRVE and fDNC are all time-independent algorithms, their performance on a given volume is the same, even if nearby volumes are omitted or shuffled in time.” We argue that, in this context, the average per-frame accuracy is relevant.

We have added a paragraph describing these additional accuracy results:

“… To visualize performance over time, we show a volume-by-volume comparison of fDNC's tracking to that of a human (Figure 3E). We also characterize model performance on a per neuron basis (Figure 3D). Finally, we used fDNC to extract whole brain calcium activity from a previously published recording of a moving animal in which two well-characterized neurons AVAL and AVAR were unambiguously labeled with an additional colored fluorophore (Hallinen et al., 2021), (Figure 3 – Supplementary Figure 1A). Calcium activity extracted from neurons AVAL and AVAR exhibited calcium activity transients when the animal underwent prolonged backward locomotion, as expected (Figure 3 – Supplementary Figure 1B).”

We have now revised language to deemphasize optogenetics and also to give more specific examples of real-time applications:

“The development of fast algorithms for tracking neurons are an important step for bringing real-time closed loop applications such as optical brain-machine interfaces (Clancy et al., 2014) and optical patch clamping (Hochbaum et al., 2014) to whole-brain imaging in freely moving animals.”

We also have added more context by comparing to the existing state of real-time methods in *C. elegans*, including for optogenetics:

“By contrast, existing real-time methods for *C. elegans* in moving animals are restricted to small subsets of neurons, are limited to two-dimensions, and work at low spatial resolution (Leifer et al., 2011; Stirman et al., 2011; Kocabas et al., 2012; Shipley et al., 2014).”

Obviously, we strive for 100% accuracy, but any automated method entails some amount of uncertainty. Our method is an important step toward improving labeling accuracy, and we believe ~80% is sufficient for many optogenetics experiments.

Nonetheless, we have broadened the discussion of real-time applications to focus less on optogenetics and to include BMI which relies only on calcium imaging.

7) What is the practical use-case for this cross-individual correspondence, since 65.8% means there are still a lot of errors. Perhaps the authors can discuss (even with some back-of-the-envelope estimates) how much this means for an experiment that compares neural activity between two different worms? What is an experiment that may require doing this fast correspondence estimation between two worms in real-time? Practically speaking, how often would one need to compute correspondences between pairs of frames between two worms? Would the overall correspondence be better if more volumes from each animal were used to find a consensus, or would the authors recommend using NeRVE in that case?

We have now added a paragraph to the discussion to clarify that correspondence is needed in two classes of use cases:

“Identifying correspondence between constellation of neurons is important for resolving two classes of problems: The first is tracking the identities of neurons across time in a moving animal. The second is mapping neurons from one individual animal onto another, and in particular onto a reference atlas, such as one obtained from electron microscopy (Witvliet et al., 2020). Mapping onto an atlas allows recordings of neurons in the laboratory to be related to known connectomic, gene expression, or other measurements in the literature”.

We have also now added four paragraphs to the discussion that put performance into a broader context, discuss potential fundamental limits, and provide one specific use case from our own work. We also now remind the reader that our model achieves 78% accuracy on the Chaudhury et al. dataset.

“The fDNC model finds neural correspondence within and across individuals with an accuracy that compares favorably to other methods. The model focuses primarily on identifying neural correspondence using position information alone. For tracking neurons within an individual using only position, fDNC achieves a high accuracy of 79%, while for across individuals using only position it achieves 64% accuracy on our dataset, and 78% on a published dataset from another group.

We expect that an upper bound may exist, set by variability introduced during the animal's development, that ultimately limits the accuracy with which any human or algorithm can find correspondence across individuals via only position information. For example, pairs of neurons in one individual that perfectly switch position with respect to another individual will never be unambiguously identified by position alone. It is unclear how close fDNC's performance of 64% on our dataset or 78% on the dataset in [2] comes to this hypothetical upper bound, but there is reason to think that at least some room for improvement remains.

Specifically, we do not expect accuracy at tracking within an individual to be fundamentally limited, in part because we do not expect two neurons to perfectly switch position on the timescale of a single recording. Therefore fDNC's 79% accuracy within-individuals suggests room for improving within-individual correspondence, and by extension, across-individual correspondence because the latter necessarily includes all of the variability of the former. One avenue for achieving higher performance could be to improve the simulator's ability to better capture variability of a real dataset, for example by using different choices of parameters in the simulator.

Even at the current level of accuracy, the ability to find correspondence across animals using position information alone remains useful. For example, we are interested in studying neural population coding of locomotion in *C. elegans* [34] , and neural correspondence at 64% accuracy will allow us to reject null hypothesis about the extent to which neural coding of locomotion is stereotyped across individuals.”

We are unable to evaluate the effect of using more volumes from each individual because we lack across-animal datasets that also have within-animal multi-volume ground truth correspondence, see Table 3.

8) "Recording" was used multiple times in the text and it's not clear whether they are time series or single volumes. For instance, it is not clear what exactly are the "12 individual animals" used for generating the training data. Are they single time frames or are they video? If videos, how many frames? It is not clear the NeuroPAL data sets are videos or single volumes.

We have now added Table 3, which lists information about the number of individuals, volumes, volume rate and other properties for each dataset used.

9) Relatedly, if many time points of 12 individual animals are used to generate training data, this is not fully synthetic. The basis of the training data from many worm heads holds a lot of information. The question is also whether all (any) of the augmentation components are necessary or useful. There should be a full characterization of the differential benefit of the different augmentations from not augmenting at all. Calling it synthetic data (e.g., line 566) may be somewhat of a misnomer.

Thank you for pointing out that the term synthetic could be confusing. To avoid ambiguity, we now use the term “semi-synthetic” throughout.

Note, however, that the 12 individual animals used by the simulator lack any ground truth correspondence within or between animals (only positions and postures are derived from measurements). We now emphasize:

“Importantly, using semi-synthetic data also allows us to train our model even when we completely lack experimentally acquired ground truth data. And indeed, in this work, semi-synthetic data is derived exclusively from measurements that lack any ground truth correspondence either within-, or across animals. All ground truth for training comes only from simulation.”

Implicit in the reviewer’s question, is another: Even if we had large numbers of ground truth datasets of multiple volumes from within a single animal, would that be sufficient to achieve good performance across animals? This is an interesting hypothetical. It is worth noting that the variability across animals is necessarily greater than the variability within animals, so it is possible that it would not be sufficient. We now mention this in the text:

“…suggests room for improving within-individual correspondence, and by extension, across-individual correspondence because the latter necessarily includes all of the variability of the former.”

One might further ask, why not collect more ground truth data? Here the transformer required O(10^5) semi-synthetic volumes to reach peak performance. It took the whole lab two weeks of dedicated effort to manually generate the ground-truth dataset with O(10^3) volumes, as described in (Nguyen et al., 2017). Based on these estimates, it would take two years to generate comparable ground truth data to train the transformer.

Reviewer #2:[…]On the first application, the true impact of fDLC may have to wait for further development of real-time cell segmentation. This seems like an imminently achievable technology. On the second application, it remains to be shown whether the 65.8% accuracy -- while quite impressive -- is sufficient to allow novel analyses and insights to be gained. For instance, if one were analyzing a dataset of 10 separately imaged worms, the overall accuracy of identifying an individual corresponding neuron among these 10 animals may be significantly lower.– Perhaps I'm being a bit nit-picky on terminology, but the use of the phrase `transfer learning` in the abstract (also in Figure 1) seems a bit of a stretch. Am I interpreting correctly that the `transfer` is between the train and test sets, without any further refinement? In what way is this `transfer learning` beyond the standard machine learning use of the test/train split?

We had sought to highlight that our test set evaluates within- and across-animal correspondence, while our semi-synthetic training set is derived from individual volumes that lack any correspondence information at all. We agree that the term transfer learning is at best confusing or at worst incorrect and have therefore removed `transfer learning’ from the text. Thank you for pointing this out.

– The methods sections mentions, in passing, that some hyper-parameter choices were made on a validation set. Which hyper-parameters were selected in this way, and what ranges of parameters were tried? In this exploration, did the authors observe that network performance was sensitive to some hyper-parameters?

As discussed in response to “Essential Revisions #2” we have added Table 7 and accompanying text describing the choice and performance of hyper-parameters.

– In the tracking results of the same worm across time, the fDLC approach treats each set of coordinates as independent measurements and does not explicitly use any temporal information. Nevertheless, I would imagine that segmented neuron positions from adjacent frames of the same movie, when the worm has not moved its pose by much, may be easier to track than pairs of frames picked at random. Is this true? What about frames that are 2, 3, etc. samples apart?

As discussed in response to “Essential Revisions #1,” we now include Figure 3E, which shows a volume by volume comparison of fDNC predictions to that of a human for each neuron over time. The fDNC algorithm does not use temporal correlations and in fact its performance on a volume is the same, even if surrounding volumes are omitted or shuffled in time.

It is interesting to ask, under what conditions would temporal information be useful? Certainly, as the review suggests, in the regime where neuron motion between frames is small compared to the mean distance between neurons, we would expect temporal information to be valuable. Any benefit of temporal information must be weighed against the potential drawback that time dependent algorithms can accumulate errors over time. In our recordings, neuron motion between frames is of similar length scale to the mean distance between closest neuron neighbors, and this may hint at why this and previous work (Nguyen et al., 2017) have been successful with time-independent strategies. We now mention this in the text:

“The recording has sufficiently large animal movement that the average distance a neuron travels between volumes (31 um) is of similar scale to the average distance between nearest neuron neighbors (35 um).“

– The acronym `fDLC` may be easily confused with some modification of DeepLabCut. While this work is also a deep learning based tracking software, I think mistaking this method for DeepLabCut may be not desirable.

We thank the reviewer for pointing this out. We have adjusted the acronym. We now use `fDNC’ for fast Deep Neural Correspondence.

Reviewer #3:This manuscript describes a deep learning model for tracking neurons in *C. elegans* worms; a side utility of the algorithm is described to be for neuron identification. The problems it is trying to address are significant as there is a need for fast neuron tracking in moving *C. elegans* whole brain imaging; the premise of the work of using synthetic data for training is interesting. The manuscript has several significant deficiencies, including claims not fully supported by evidence and overreaching conclusions.Major strengths:1. The idea of using augmentation to real data to generate training sets for ML model is interesting, particularly in situations where data are hard to come by.2. fDLC's speed is attractive for the use cases.Major weaknesses:1. For tracking to have ~80% accuracy is not meaningful. First, this accuracy is an average, and it has no bearing on whether a cell can be *continuously* tracked. The traces may be broken, and worse, wrong cells are linked together. This 80% does not guarantee anything at this point. Having an accuracy on per-frame basis is not useful at all. To actually have an impact on tracking, traces have to be shown, and these traces need to be verified. There are existing data from the authors themselves and others. One would have to show that the traces are similar, and better yet, the temporal PCs are similar. The tracking having 80% accuracy cannot be used for optogenetics at all. It is not meaningful to fire the laser at cells with 20% uncertainty in their identities and carry out any meaningful experiments. This claim does not make sense. The text on optogenetics needs to be significantly toned down, or better yet, removed.

Please see detailed response to “Essential Revisions: #6”.

2. What the Transformer network learned with the data is unclear. The paper does not show exactly what the Transformer network has learned – what features of the data are important? This is critical, as it is possible that another form of information is actually being learned from the data. For instance, in training where the hand-curated cells are used, the cells may be entered in a particular order. It is therefore possible that the Transformer network is learning the order of which the cells are entered, rather than the actual spatial relationships. To show that the Transformer network is really learning something meaningful in the data, one would have to scramble the order of the data and show that the results are not different.

The order of data is indeed scrambled in both training and test sets and we have clarified this in the text. Therefore the model is not learning the order. Please see detailed response to “Essential Revisions: #1”.

3. The authors stressed that the learning does not require users to prescribe what to look for, but the warping, transformation, noises added are in essence adding information in user-defined way. This claim does not make sense. In the text, the authors also use language such as "roughly matched (their) estimate of variability observed by eye". This is not rigorous and seems dangerous. Exact details and rationales of choices for the warping, transformation, noises added, etc need to be included and fully justified.

We have now removed that text and now specify in the discussion that one avenue for future improvement is to better tune the simulator to capture variability. “One avenue for achieving higher performance could be to improve the simulator's ability to better capture variability of a real dataset, for example by using different choices of parameters in the simulator.”

4. Clarity of algorithm performance is lacking. For instance, there are no training curves shown for the algorithm.

Training curves have been added in Figure 2 – Supplementary Figure 1.

5. Related, importantly, the accuracy of the algorithm must be very much data-dependent. Sources that can perturb a perfect scenario need to be examined. For instance, how would cells' activities in GCaMP recordings affect accuracy? How would segmentation error affect accuracy? It is not possible to evaluate the real-world utility if these issues are not explored. For all we know, it could be the best data that are fed to the algorithm that is used to calculate the accuracies here.

To account for differences in data, and to provide fair comparison against other methods, we evaluate performance on multiple recordings from our own group including those that we have published previously, and on all recordings in a published dataset from a different group (Chaudhary, 2021). Performance on individual recordings in each dataset are visible in Figure 4, and they span a wide range. Also our model performs better on the Chaudhary dataset (78.2%) than our own (64.1%). That we use a wide range of datasets, and that our model performs even better on another group’s dataset is evidence that we are not using only “the best data to calculate accuracies.”

Regarding GCaMP, we note that accuracy reported in Figure 3 is evaluated on a recording that contains GCaMP activity (originally from Nguyen et al., 2017). Moreover, we also have now added a new example where we apply our method to a recording we recently published (Hallinen et al., 2021) and in this case we also show that GCaMP activity behaves as expected, Figure 3 – Supplementary Figure 1.

6. The authors stated that there is a trade-off between accuracy and coverage. This is an important point, but the authors did not fully characterize such trade-off (related to the accuracy comment above); nor was the coverage assumption/definition that went into each part of the work clearly stated. In the tracking part, what would the coverage be? How is it defined? Comparisons to literature algorithm for neuron identification is should not be done when the coverage is also not well defined, i.e. the denominators for the percentages in table 4 are ill-defined.

Regarding coverage, it is important to note that the algorithm assigns every segmented neuron in the test or template (whichever has fewer) a match. We now reiterate this point more often in the text:

“Every segmented neuron in the test or template (whichever has fewer) is assigned a match. Accuracy is defined as the number of proposed matches that agree with ground truth, divided by the total number of ground truth matches. The number of ground truth matches is a property of the dataset used to evaluate our model and is listed in Table 3.”

The numerator and denominator are now well defined for all calculations of the accuracy of our model, NeRVE and CPD, including in Table 4(Table 2 in new version): “number of proposed matches that agree with ground truth, divided by the total number of ground truth matches”.

We now list information about the denominator explicitly in Table 3 by showing the ground truth matches from test and template pairs sampled from each dataset. We note that this is a property of the dataset and not of the model. Now that we have a more simplified definition of accuracy, a discussion of “coverage” is no longer relevant and has been removed.

There remains the question of how best to compare our model’s accuracy to that of the CRF model from (Chaudhary et al) because that model reports accuracy using templates that are privileged (see below). We have added two paragraphs describing the specific assumption under which our two models can be directly compared:

“The CRF model compares test and template like we do, but their template is privileged in the sense that it is derived from either the literature (“open atlas”) or aggregated from their other recordings (``data-driven''). Moreover, the data driven atlas incorporates statistics about variability from across their recordings. By contrast our template is simply one of the other recordings in the dataset. Pairing a test with a privileged template provides slightly more ground truth matches on which to evaluate performance, because the privileged template contains more ground truth labels. We expect the difference is modest, however, because the number of ground truth matches is still limited by the number of neurons with ground truth labels in the test.

Nonetheless, we must make an assumption to directly compare reported accuracy of the CRF model on the dataset in (Chaudhary et al) to the fDNC model's performance on the same dataset. We must assume that on average there is nothing particularly special about those neurons that have ground truth labels present in the intersection of test and privileged template, but that lack a ground truth label in a non-privileged template sampled from the recordings. Under this assumption, we compared the reported performance of the CRF model on the published dataset in (Chaudhary et al) to the performance of the fDNC model evaluated on the same dataset (Table 2).”

7. Clarity of the experimental data is lacking. "Recording" was used multiple times in the text and it's not clear whether they are time series or single volumes. For instance, it is not clear what exactly are the "12 individual animals" used for generating the training data. Are they single time frames or are they video? If videos, how many frames? It is not clear the NeuroPAL data sets are videos or single volumes.

See response to “Essential Revisions #8”.

8. Related to the issue above, if many time points of 12 individual animals are used to generate training data, this is not at all synthetic. The basis of the training data from many worm heads holds a lot of information. The question is also whether all (any) of the augmentation components are necessary or useful. There should be a full characterization of the differential benefit of the different augmentations from not augmenting at all. Calling it synthetic data in my opinion is a misnomer (e.g. line 566).

See response to “Essential Revisions #9”.

9. Generally speaking, if the algorithm is very generalizable and extremely fast, and quite accurate as the authors claim, it should be fairly simple to use it on a real whole-brain experiment data set and show that meaningful conclusions can come of it. Without this, one should not make such claims that "The method is fast and predicts correspondence in 10 ms making it suitable for future real-time applications."

See response to “Essential Revisions #5”.

[Editors' note: further revisions were suggested prior to acceptance, as described below.]

Essential Revisions:1. The data are from the authors themselves, not peer-reviewed, and not independently validated. The authors did not use, for instance, the Zimmer lab's data; the reason why was unclear to the reviewers. Also, the authors themselves have at least one volume of sensible data from their own previous work (NeRVE, Nguyen et al., 2017) in which they actually performed PCA on the GCaMP data. Applying fDNC to that set of data and showing that PCAs are comparable would make their claim much stronger.

– The published population recordings that we know of from the Zimmer group are for immobilized animals. Tracking an immobile recording would not be a good demonstration of the fDNC method. Neuron configurations do not change over time in immobilized animals, so tracking during immobilization is relatively trivial. fDNC would only be used for tracking neurons in moving animals.

– Figure 3e shows fDNC applied to the requested GCaMP recording from Nguyen et al. 2017.

We have further clarified the caption to make this clear.

“Detailed comparison of fdNC tracking to human annotation of a moving GCaMP recording from Nguyen et al. (2017) [18]”.

– The most stringent comparison we can perform is to compare fDNC tracking to human ground truth tracking, as we have done in Figure 3d and e with the GCaMP recordings from Nguyen 2017. Comparing calcium activity, as suggested, is one step further removed, and is a less informative comparison. If the reviewer’s goal is to assess whether fDNC-tracked neurons can result in plausible calcium traces, Figure 3—figure supplement 1 (p. 27) shows that it does.

– We disagree that PCA analysis of neural activity is informative or relevant to demonstrate the method and including the analysis risks dragging the paper in a confusing new direction. But to facilitate peer review, we have included the requested analysis below. Left shows neural activity from Ngyuen et al. 2017 using human tracking projected into its first two principal components. Right shows activity from the same recording tracked via fDNC and projected into *its* first two principal components.

Comparing tracking via calcium activity in this way is less stringent and less informative than comparing tracking directly as in Figure 3d and e. We also find these plots difficult to interpret and potentially confusing. Finally, far from being a standard analysis, in published work low dimensional neural state space trajectories have only previously been applied to immobile *C. elegans*, not to moving animals. So including these plots would also broach new scientific ground that is beyond the scope of this methods paper.

2. The accuracy is a central claim in the paper. It is good that the authors now define what accuracy is in the text, but it is still confusing. A match between the template and the test does not assign a name necessarily -- unless the template neurons already have labels/identities from the ground truth information. From the text, it seems that the template is used as the reference with identities already assigned and that only the neurons common to both the test and template are considered since the denominator of the accuracy is defined as "the total number of ground truth matches". (Another interpretation of the definition would suggest that neurons that both the template and the test got wrong but matches each other would have been counted as accurate?!)

There may be a misunderstanding. Ground truth labels are only used for evaluating performance after the fact, they are not part of the model. See response to Essential feedback #3. All neurons get matched, no label is required. As stated in the text,

“Every segmented neuron in the test or template (whichever has fewer) is assigned a match.”

There are two issues – the definition is not applicable for some other methods and that this definition is artificially favorable for fDNC.a. In Table 2, the authors compare the accuracies of fDNC to that of CPD and CRF (ref 3). This is not appropriate. fDNC and CPD both use template matching, while CRF does not. This is to say that the accuracy definition is not the same for these methods.

We have rewritten the section where we compare fDNC to CPD to highlight the reviewer’s point about template matching and the differences in the models between fDNC and CRF. We now make explicit the assumptions under which we compare fDNC and CRF, despite their differences, and provide quantitative bounds on the range of possible assumptions. And, out of an abundance of caution, we have tempered our conclusions about relative accuracy. We say that fDNC’s accuracy is “comparable” to CRF in addition to having other advantages.

We hope the reviewers and editors will recognize the value in comparing methods on the same published datasets, and understand that in this case an assumption is necessary to make the comparison.

“We further sought to compare the fDNC model to the reported accuracy of a recent model called Conditional Random Fields (CRF) from (Chaudhary et. al, 2021) by evaluating fDNC on the same published dataset from that work. […] Taken together, we conclude that the fDNC model's accuracy is comparable to that of the CRF model while also providing other advantages.”

b. The accuracy of fDNC is artificially more favorable. NeuroPAL datasets do not reliably identify the same neurons. When using one NeuroPAL dataset as template, and another as the test set, the matches are on the order of 70-80%. The definition of accuracy the authors use, therefore, is artificially high (by some significant percentage). The errors associated in neurons not common to the test and the template are discounted.

– For neurons that are not part of the set of ground truth labels that intersect test and template, we neither catch errors nor catch correct matches. It is not obvious to us whether this undercounts or overcounts our accuracy compared to the hypothetical in which a human had a complete set of ground truth labels at their disposal.

– We have carefully considered alternative definitions of accuracy and of all of them, this definition best reflects the information we have. We note that (Chaudhury et al.) faces the same challenge in their framework with respect to neurons that lack ground truth in their test worms and they approach this similarly. They evaluate performance on only those neurons with ground truth labels in the test and ignore segmented neurons that lack ground truth labels for the purposes of reporting accuracy.

– In Table 3 we provide quantitative details about the number of segmented neurons per individual, the number of ground truth labels per individual, and the number of ground truth matches per pair, so that a reader has all of the information they need to understand the ramifications of our choice of accuracy.

c. The coverage and the accuracy discussion should be restored.

The key points about how we define accuracy and how we think about the denominator are present and clearer than in the initial submission. Table 3 in particular, precisely quantifies how many neurons are segmented and how many have ground truth labels. The previous round of reviewer feedback made clear that the “coverage and accuracy” framing was causing confusion. We hesitate to revive it.

3. Implying that fDNC is not "data-privileged" is false (page 13). fDNC is not naive – information from 4000 volumes from 12 animals is there, and fDNC must use a known annotated NeuroPAL dataset as a template, and therefore there is information again (e.g. variability of positions etc). Revising the discussion around this point is important.

– We rewrote the section (pasted above, in response to Reviewer #3 feedback 2b.) and removed the word “privileged” as it is imprecise and may be causing confusion. Thank you for pointing this out.

– There may be a misunderstanding. fDNC finds matches between two configurations. It does not require a known annotated NeuroPAL dataset as a template (and does not use NeuroPAL for training). We added new text to clarify:

“Later in the work we use ground truth information from human annotated NeuroPAL (Yemini et al., 2021) strains to evaluate the performance of our model, but no NeuroPAL strains were used for training.”

– For example, in Figure 3 fDNC finds correspondence between two worms even though it is blind to any NeuroPAL color information. NeuroPAL is then used only to evaluate the performance of the matches. We added text to clarify:

“NeuroPAL worms contain extra color information that allows a human to assign ground truth labels to evaluate the model's performance. Crucially, the fDNC model was blinded to this additional color information. In these experiments, NeuroPAL color information was only used to evaluate performance after the fact, not to find correspondence.”